# Clustering with Noisy Queries

**Arya Mazumdar and Barna Saha**
College of Information and Computer Sciences
University of Massachusetts Amherst
Amherst, MA 01003
{arya,barna}@cs.umass.edu

## Abstract

In this paper, we provide a rigorous theoretical study of clustering with noisy queries. Given a set of $n$ elements, our goal is to recover the true clustering by asking minimum number of pairwise queries to an oracle. Oracle can answer queries of the form "do elements $u$ and $v$ belong to the same cluster?"-the queries can be asked interactively (adaptive queries), or non-adaptively up-front, but its answer can be erroneous with probability $p$. In this paper, we provide the first information theoretic lower bound on the number of queries for clustering with noisy oracle in both situations. We design novel algorithms that closely match this query complexity lower bound, even when the number of clusters is unknown. Moreover, we design computationally efficient algorithms both for the adaptive and non-adaptive settings. The problem captures/generalizes multiple application scenarios. It is directly motivated by the growing body of work that use crowdsourcing for *entity resolution*, a fundamental and challenging data mining task aimed to identify all records in a database referring to the same entity. Here crowd represents the noisy oracle, and the number of queries directly relates to the cost of crowdsourcing. Another application comes from the problem of *sign edge prediction* in social network, where social interactions can be both positive and negative, and one must identify the sign of all pair-wise interactions by querying a few pairs. Furthermore, clustering with noisy oracle is intimately connected to correlation clustering, leading to improvement therein. Finally, it introduces a new direction of study in the popular *stochastic block model* where one has an incomplete stochastic block model matrix to recover the clusters.

## 1 Introduction

Clustering is one of the most fundamental and popular methods for data classification. In this paper we initiate a rigorous theoretical study of clustering with the help of a noisy oracle, a model that captures many application scenarios and has drawn significant attention in recent years.

Suppose we are given a set of $n$ points, that need to be clustered into $k$ clusters where $k$ is unknown to us. Suppose there is an oracle that can answer pair-wise queries of the form, "do $u$ and $v$ belong to the same cluster?". Repeating the same question to the oracle always returns the same answer, but the answer could be wrong with probability $p = \frac{1}{2} - \lambda, \lambda > 0$ (i.e., slightly better than random answer). We are interested to find the minimum number of queries needed to recover the true clusters with high probability. Understanding query complexity of the noisy oracle model is a fundamental theoretical question [25] with many existing works on sorting and selection [7, 8] where queries are erroneous with probability $p$, and repeating the same question does not change the answer. Here we study the basic clustering problem under this setting which also captures several fundamental applications.

**Crowdsourced Entity Resolution.** Entity resolution (ER) is an important data mining task that tries to identify all records in a database that refer to the same underlying entity. Starting with the

seminal work of Fellegi and Sunter [26], numerous algorithms with variety of techniques have been developed for ER [24, 28, 40, 19]. Still, due to ambiguity in representation and poor data quality, accuracy of automated ER techniques has been unsatisfactory. To remedy this, a recent trend in ER has been to use human in the loop. In this setting, humans are asked simple pair-wise queries adaptively, "do $u$ and $v$ represent the same entity?", and these answers are used to improve the final accuracy [30, 54, 56, 27, 52, 21, 29, 37, 55, 46]. Proliferation of crowdsourcing platforms like Amazon Mechanical Turk (AMT), CrowdFlower etc. allows for easy implementation. However, data collected from non-expert workers on crowdsourcing platforms are inevitably noisy. A simple scheme to reduce errors could be to take a majority vote after asking the same question to multiple independent crowd workers. However, often that is not sufficient. Our experiments on several real datasets (see experimentation section for details) with answers collected from AMT [31, 52] show majority voting could even increase the errors. Interestingly, such an observation has been made by a recent paper as well [51]. There are more complex querying model [51, 55, 53], and involved heuristics [31, 52] to handle errors in this scenario. Let $p, 0 < p < 1/2$, be the probability of error[1] of a query answer which might also be the aggregated answer after repeating the query several times. Therefore, once the answer has been aggregated, it cannot change. In all crowdsourcing works, the goal is to minimize the number of queries to reduce the cost and time of crowdsourcing, and recover the entities (clusters). This is exactly clustering with noisy oracle. While several heuristics have been developed [52, 30, 53], here we provide a rigorous theory with near-optimal algorithms and hardness bounds.

Another recent work that is conceptually close is by Asthiani et al. [4], where pair-wise queries are used for clustering. However, the setting is very different. They consider the specific NP-hard $k$-means objective with distance matrix which must be a metric and must satisfy a deterministic separation property.

**Signed Edge Prediction.** The edge sign prediction problem can be defined as follows. Suppose we are given a social network with signs on all its edges, but the sign from node $u$ to $v$, denoted by $s(u, v) \in \{\pm 1\}$ is hidden. The goal is to recover these signs as best as possible using minimal amount of information. Social interactions or sentiments can be both positive ("like", "trust") and negative ("dislike", "distrust"). [41] provides several such examples; e.g., Wikipedia, where one can vote for or against the nomination of others to adminship [10], or Epinions and Slashdots where users can express trust or distrust, or can declare others to be friends or foes [9, 39]. Initiated by [11, 34], many techniques and related models using convex optimization, low-rank approximation and learning theoretic approaches have been used for this problem [17, 12, 14]. Recently [16, 14, 48] proposed the following model for edge sign prediction. We can query a pair of nodes $(u, v)$ to test whether $s(u, v) = +1$ indicating $u$ and $v$ belong to the same cluster or $s(u, v) = -1$ indicating they are not. However, the query fails to return the correct answer with probability $0 < p < 1/2$, and we want to query the minimal possible pairs. This is exactly the case of *clustering with noisy oracle*. Our result significantly improves, and generalizes over [16, 14, 48].

**Correlation Clustering.** In fact, when all pair-wise queries are given, and the goal is to recover the maximum likelihood (ML) clustering, then our problem is equivalent to *noisy correlation clustering* [6, 44]. Introduced by [6], correlation clustering is an extremely well-studied model of clustering. We are given a graph $G = (V, E)$ with each edge $e \in E$ labelled either $+1$ or $-1$, the goal of correlation clustering is to either (a) minimize the number of disagreements, that is the number of intra-cluster $-1$ edges and inter-cluster $+1$ edges, or (b) maximize the number of agreements that is the number of intra-cluster $+1$ edges and inter-cluster $-1$ edges. Correlation clustering is NP-hard, but can be approximated well with provable guarantees [6]. In a random noise model, also introduced by [6] and studied further by [44], we start with a ground truth clustering, and then each edge label is flipped with probability $p$. This is exactly the graph we observe if we make all possible pair-wise queries, and the ML decoding coincides with correlation clustering. The proposed algorithm of [6] can recover in this case all clusters of size $\omega(\sqrt{|V| \log |V|})$, and if "all" the clusters have size $\Omega(\sqrt{|V|})$, then they can be recovered by [44]. Using our proposed algorithms for clustering with noisy oracle, we can also recover significantly smaller sized clusters given the number of clusters are not too many. Such a result is possible to obtain using the repeated-peeling technique of [3]. However, our running time is significantly better. E.g. for $k \leq n^{1/6}$, we have a running time of $O(n \log n)$, whereas for [3], it is dominated by the time to solve a convex optimization over $n$-vertex graph which is at least $O(n^3)$.

**Stochastic Block Model (SBM).** The clustering with faulty oracle is intimately connected with the *planted partition model*, also known as the stochastic block model [36, 23, 22, 2, 1, 32, 18, 49]. The stochastic block model is an extremely well-studied model of random graphs where two vertices within the same community share an edge with probability $p'$, and two vertices in different communities share an edge with probability $q'$. It is often assumed that $k$, the number of communities, is a constant (e.g. $k = 2$ is known as the *planted bisection model* and is studied extensively [1, 49, 23] or a slowly growing function of $n$ (e.g. $k = o(\log n)$). There are extensive literature on characterizing the threshold phenomenon in SBM in terms of the gap between $p'$ and $q'^2$ (e.g. see [2] and therein for many references) for exact and approximate recovery of clusters of nearly equal size. If we allow for different probability of errors for pairs of elements based on whether they belong to the same cluster or not, then the resultant faulty oracle model is an intriguing generalization of SBM. Consider the probability of error for a query on $(u, v)$ is $1 - p'$ if $u$ and $v$ belong to the same cluster and $q'$ otherwise; but now, we can only learn a subset of the entries of an SBM matrix by querying adaptively. Understanding how the threshold of recovery changes for such an "incomplete" or "space-efficient" SBM will be a fascinating direction to pursue. In fact, our lower bound results extend to asymmetric probability values, while designing efficient algorithms and sharp thresholds are ongoing works. In [15], a locality model where measurements can only be obtained for nearby nodes is studied for two clusters with non-adaptive querying and allowing repetitions. It would also be interesting to extend our work with such locality constraints.

In a companion paper, we have studied a related problem where the queries are not noisy and certain similarity values between each pair of elements are available [47]. Most of the results of the two papers are available online in a more extensive version [45].

**Contributions.** Formally the *clustering with a noisy oracle* is defined as follows.

**Problem** (Query-Cluster ). *Consider a set of points $V \equiv [n]$ containing $k$ latent clusters $V_i$, $i = 1, \ldots, k$, $V_i \cap V_j = \emptyset$, where $k$ and the subsets $V_i \subseteq [n]$ are unknown. There is an oracle $\mathcal{O}_{p,q} : V \times V \to \{\pm 1\}$, with two error parameters $p, q : 0 < p < q < 1$. The oracle takes as input a pair of vertices $u, v \in V \times V$, and if $u, v$ belong to the same cluster then $\mathcal{O}_{p,q}(u, v) = +1$ with probability $1 - p$ and $\mathcal{O}_{p,q}(u, v) = -1$ with probability $p$. On the other hand, if $u, v$ do not belong to the same cluster then $\mathcal{O}_{p,q}(u, v) = +1$ with probability $1 - q$ and $\mathcal{O}_{p,q}(u, v) = -1$ with probability $q$. Such an oracle is called a* binary asymmetric channel. *A special case would be when $p = 1 - q = \frac{1}{2} - \lambda, \lambda > 0$, the binary* symmetric *channel, where the error rate is the same $p$ for all pairs. Except for the lower bound, we focus on the symmetric case in this paper. Note that the oracle returns the same answer on repetition. Now, given $V$, find $Q \subseteq V \times V$ such that $|Q|$ is minimum, and from the oracle answers it is possible to recover $V_i$, $i = 1, 2, ..., k$ with high probability[3]. Note that the entries of $Q$ can be chosen adaptively based on the answers of previously chosen queries.*

Our contributions are as follows.

• *Lower Bound (Section 2).* We show that $\Omega(\frac{nk}{\Delta(p\|q)})$ is the information theoretic lower bound on the number of adaptive queries required to obtain the correct clustering with high probability even when the clusters are of similar size (see, Theorem 1). Here $\Delta(p\|q)$ is the Jensen-Shannon divergence between Bernoulli $p$ and $q$ distributions. For the symmetric case, that is when $p = 1 - q$, $\Delta(p\|1 - p) = (1 - 2p) \log \frac{1-p}{p}$. In particular, if $p = \frac{1}{2} - \lambda$, our lower bound on query complexity is $\Omega(\frac{nk}{\lambda^2}) = \Omega(\frac{nk}{(1-2p)^2})$. Developing lower bounds in the interactive setting especially with noisy answers appears to be significantly challenging as popular techniques based on Fano-type inequalities for multiple hypothesis testing [13, 42] do not apply, and we believe our technique will be useful in other noisy interactive learning settings.

• *Information-Theoretic Optimal Algorithm (Section 3 and B.1).* For the symmetric error case, we design an algorithm which asks at most $O(\frac{nk \log n}{(1-2p)^2})$ queries (Theorem 2) matching the lower bound within an $O(\log n)$ factor, whenever $p = \frac{1}{2} - \lambda$.

• *Computationally Efficient Algorithm (Section 3.2 and B.2).* We next design an algorithm that is computationally efficient and runs in $O(nk \log n + k^{1+2\omega})$ time where $\omega \leq 2.373$ is the exponent of fast matrix multiplication and asks at most $O(nk \log(n) + \min(nk^2 \log(n), k^5 \log^2 n))$ queries treating $p$ as a constant[4]. Note that most prior works in SBM, or works on edge sign detection, only

consider the case when $k$ is a constant [2, 32, 18], even just $k = 2$ [49, 1, 16, 14, 48]. For small values of $k$, we get a highly efficient algorithm. We can use this algorithm to recover all clusters of size at least $\min(k, \sqrt{n}) \log n$ for correlation clustering on noisy graph, improving upon the results of [6, 44]. As long as $k = o(\sqrt{n})$, this improves upon the running time of $O(n^3)$ in [3].

• *Nonadaptive Algorithm (Section B.3).* When the queries must be done up-front, for $k = 2$, we give a simple $O(n \log n)$ time algorithm that asks $O(\frac{n \log n}{(1-2p)^4})$ queries improving upon [48] where a polynomial time algorithm (at least with a running time of $O(n^3)$) is shown with number of queries $O(n \log n / (1/2 - p)^{\frac{\log n}{\log \log n}})$ and over [16, 14] where $O(n\text{poly} \log n)$ queries are required under certain conditions on the clusters. Our result generalizes to $k > 2$, and we show interesting lower bounds in this setting (Appendix C in the supplementary material). Further, we derive new lower bounds showing trade-off between queries and threshold of recovery for incomplete SBM in Appendix C.

## 2 Lower bound for the faulty-oracle model

Note that we are not allowed to ask the same question multiple times to get the correct answer. In this case, even for probabilistic recovery, a minimum size bound on cluster size is required. For example, consider the following two different clusterings. $C_1 : V = \sqcup_{i=1}^{k-2} V_i \sqcup \{v_1, v_2\} \sqcup \{v_3\}$ and $C_2 : V = \sqcup_{i=1}^{k-2} V_i \sqcup \{v_1\} \sqcup \{v_2, v_3\}$. Now if one of these two clusterings are given to us uniformly at random, no matter how many queries we do, we will fail to recover the correct clustering with positive probability. Therefore, the challenge in proving lower bounds is when clusters all have size more than a minimum threshold, or when they are all nearly balanced. This removes the constraint on the algorithm designer on how many times a cluster can be queried with a vertex and the algorithms can have greater flexibility. Our lower bound holds for a large set of clustering instances. We define a clustering to be *balanced* if either of the following two conditions hold 1) the minimum size of a cluster is $\geq \frac{n}{20k}$, 2) the maximum size of a cluster is $\leq \frac{4n}{k}$. For any balanced clustering, we prove a lower bound on the number of queries required.

Our main lower bound in this section uses the Jensen-Shannon (JS) divergence. The well-known KL divergence is defined between two probability mass functions $f$ and $g$: $D(f\|g) = \sum_i f(i) \log \frac{f(i)}{g(i)}$. Further define the JS divergence as: $\Delta(f\|g) = \frac{1}{2}(D(f\|g) + D(g\|f))$. In particular, the KL and JS divergences between two Bernoulli random variable with parameters $p$ and $q$ are denoted with $D(p\|q)$ and $\Delta(p\|q)$ respectively.

**Theorem 1** (Query-Cluster Lower Bound). *For any balanced clustering instance, if any (randomized) algorithm does not make $\Omega\left(\frac{nk}{\Delta(p\|q)}\right)$ expected number of queries then the recovery will be incorrect with probability at least $0.29 - O(\frac{1}{k})$.*

Note that the lower bound is more effective when $p$ and $q$ are close. Moreover our actual lower bound is slightly tighter with the expected number of queries required given by $\Omega\left(\frac{nk}{\min\{D(q\|p), D(p\|q)\}}\right)$.

**Proof of Theorem 1.** We have $V$ to be the $n$-element set to be clustered: $V = \sqcup_{i=1}^{k} V_i$. To prove Theorem 1 we first show that, if the number of queries is small, then there exist $\Omega(k)$ number of clusters, that are not being sufficiently queried with. Then we show that, since the size of the clusters cannot be too large or too small, there exists a decent number of vertices in these clusters.

The main piece of the proof of Theorem 1 is Lemma 1. We provide a sketch of this lemma here, the full proof, which is inspired by a technique of lower bounding regret in multi-arm bandit problems (see [5, 38]) is given in Appendix A in the supplementary material.

**Lemma 1.** *Suppose, there are $k$ clusters. There exist at least $\frac{4k}{5}$ clusters such that for each element $v$ from any of these clusters, $v$ will be assigned to a wrong cluster by any randomized algorithm with probability $0.29 - 10/k$ unless the total number of queries involving $v$ is more than $\frac{k}{10\Delta(p\|q)}$.*

*Proof-sketch of Lemma 1.* Let us assume that the $k$ clusters are already formed, and all elements except for one element $v$ has already been assigned to a cluster. Note that, queries that do not involve $v$ plays no role in this stage.

Now the problem reduces to a hypothesis testing problem where the $i$th hypothesis $H_i$ for $i = 1, \ldots, k$, denotes that the true cluster for $v$ is $V_i$. We can also add a null-hypothesis $H_0$ that stands for the vertex belonging to none of the clusters (hypothetical). Let $P_i$ denote the joint probability distribution of our observations (the answers to the queries involving vertex $v$) when $H_i$ is true, $i = 1, \ldots, k$. That is for any event $\mathcal{A}$ we have $P_i(\mathcal{A}) = \Pr(\mathcal{A}|H_i)$.

Suppose $T$ denotes the total number of queries made by an (possibly randomized) algorithm at this stage before assigning a cluster. Let the random variable $T_i$ denote the number of queries involving cluster $V_i, i = 1, \ldots, k$. In the second step, we need to identify a set of clusters that are not being queried with enough by the algorithm.

We must have, $\sum_{i=1}^{k} \mathbb{E}_0 T_i = T$. Let $J_1 \equiv \{i \in \{1, \ldots, k\} : \mathbb{E}_0 T_i \leq \frac{10T}{k}\}$. That is $J_1$ contains clusters which were involved in less than $\frac{10T}{k}$ queries before assignment. Let $\mathcal{E}_i \equiv$ {the algorithm outputs cluster $V_i$} and $J_2 = \{i \in \{1, \ldots, n\} : P_0(\mathcal{E}_i) \leq \frac{10}{k}\}$. The set of clusters, $J = J_1 \cap J_2$ has size, $|J| \geq 2 \cdot \frac{9k}{10} - k = \frac{4k}{5}$.

Now let us assume that we are given an element $v \in V_j$ for some $j \in J$ to cluster ($H_j$ is the true hypothesis). The probability of correct clustering is $P_j(\mathcal{E}_j)$. In the last step, we give an upper bound on probability of correct assignment for this element.

We must have, $P_j(\mathcal{E}_j) = P_0(\mathcal{E}_j) + P_j(\mathcal{E}_j) - P_0(\mathcal{E}_j) \leq \frac{10}{k} + |P_0(\mathcal{E}_j) - P_j(\mathcal{E}_j)| \leq \frac{10}{k} + \|P_0 - P_j\|_{TV} \leq \frac{10}{k} + \sqrt{\frac{1}{2}D(P_0\|P_j)}$. where $\|P_0 - P_j\|_{TV}$ denotes the total variation distance between two distributions and and in the last step we have used the relation between total variation and divergence (Pinsker's inequality). Since $P_0$ and $P_j$ are the joint distributions of the independent random variables (answers to queries) that are identical to one of two Bernoulli random variables: $Y$, which is Bernoulli($p$), or $Z$, which is Bernoulli($q$), it is possible to show, $D(P_0\|P_j) \leq \frac{10T}{k} D(q\|p)$.

Now plugging this in,

$$P_j(\mathcal{E}_j) \leq \frac{10}{k} + \sqrt{\frac{1}{2}\frac{10T}{k}D(q\|p)} \leq \frac{10}{k} + \sqrt{\frac{1}{2}} = \frac{10}{k} + 0.707,$$

if $T \leq \frac{k}{10D(q\|p)}$. Had we bounded the total variation distance with $D(P_j\|P_0)$ in the Pinsker's inequality then we would have $D(p\|q)$ in the denominator. $\square$

Now we are ready to prove Theorem 1.

*Proof of Theorem 1.* We will show the claim by considering a balanced input. Recall that for a balanced input either the maximum size of a cluster is $\leq \frac{4n}{k}$ or the minimum size of a cluster is $\geq \frac{n}{20k}$. We will consider the two cases separately for the proof.

*Case 1: the maximum size of a cluster is $\leq \frac{4n}{k}$.*

Suppose, the total number of queries is $T'$. That means number of vertices involved in the queries is $\leq 2T'$. Note that there are $k$ clusters and $n$ elements. Let $U$ be the set of vertices that are involved in less than $\frac{16T'}{n}$ queries. Clearly, $(n - |U|)\frac{16T'}{n} \leq 2T'$, or $|U| \geq \frac{7n}{8}$.

Now we know from Lemma 1 that there exists $\frac{4k}{5}$ clusters such that a vertex $v$ from any one of these clusters will be assigned to a wrong cluster by any randomized algorithm with probability $1/4$ unless the expected number of queries involving this vertex is more than $\frac{k}{10\Delta(q\|p)}$.

We claim that $U$ must have an intersection with at least one of these $\frac{4k}{5}$ clusters. If not, then more than $\frac{7n}{8}$ vertices must belong to less than $k - \frac{4k}{5} = \frac{k}{5}$ clusters. Or the maximum size of a cluster will be $\frac{7n \cdot 5}{8k} > \frac{4n}{k}$, which is prohibited according to our assumption.

Now each vertex in the intersection of $U$ and the $\frac{4k}{5}$ clusters are going to be assigned to an incorrect cluster with positive probability if, $\frac{16T'}{n} \leq \frac{k}{10\Delta(p\|q)}$. Therefore we must have $T' \geq \frac{nk}{160\Delta(p\|q)}$.

*Case 2: the minimum size of a cluster is $\geq \frac{n}{20k}$.*

Let $U'$ be the set of clusters that are involved in at most $\frac{16T'}{k}$ queries. That means, $(k - |U'|)\frac{16T'}{k} \leq 2T'$. This implies, $|U'| \geq \frac{7k}{8}$. Now we know from Lemma 1 that there exist $\frac{4k}{5}$ clusters (say $U^*$) such

that a vertex $v$ from any one of these clusters will be assigned to a wrong cluster by any randomized algorithm with probability $1/4$ unless the expected number of queries involving this vertex is more than $\frac{k}{10\Delta(p\|q)}$. Quite clearly $|U^* \cap U| \geq \frac{7k}{8} + \frac{4k}{5} - k = \frac{27k}{40}$.

Consider a cluster $V_i$ such that $i \in U^* \cap U$, which is always possible because the intersection is nonempty. $V_i$ is involved in at most $\frac{16T'}{k}$ queries. Let the minimum size of any cluster be $t$. Now, at least half of the vertices of $V_i$ must each be involved in at most $\frac{32T'}{kt}$ queries. Now each of these vertices must be involved in at least $\frac{k}{10\Delta(p\|q)}$ queries (see Lemma 1) to avoid being assigned to a wrong cluster with positive probability. This means $\frac{32T'}{kt} \geq \frac{k}{10\Delta(p\|q)}$ or $T' = \Omega\left(\frac{nk}{\Delta(p\|q)}\right)$, since $t \geq \frac{n}{20k}$. $\qquad\square$

# 3 Algorithms

Let $V = \sqcup_{i=1}^{k} V_i$ be the true clustering and $V = \sqcup_{i=1}^{k} \hat{V}_i$ be the maximum likelihood (ML) estimate of the clustering that can be found when all $\binom{n}{2}$ queries have been made to the faulty oracle. Our first result obtains a query complexity upper bound within an $O(\log n)$ factor of the information theoretic lower bound. The algorithm runs in quasi-polynomial time, and we show this is the optimal possible assuming the famous *planted clique* hardness. Next, we show how the ideas can be extended to make it computationally efficient. We consider both the adaptive and non-adaptive versions. The missing proofs and details are provided in Appendix B in the supplementary document.

## 3.1 Information-Theoretic Optimal Algorithm

In particular, we prove the following theorem.

**Theorem 2.** *There exists an algorithm with query complexity $O(\frac{nk \log n}{(1-2p)^2})$ for* Query-Cluster *that returns the ML estimate with high probability when query answers are incorrect with probability $p < \frac{1}{2}$. Moreover, the algorithm returns all true clusters of $V$ of size at least $\frac{C \log n}{(1-2p)^2}$ for a suitable constant $C$ with probability $1 - o_n(1)$.*

**Remark 1.** *Assuming $p = \frac{1}{2} - \lambda$, as $\lambda \to 0$, $\Delta(p\|1-p) = (1-2p)\ln\frac{1-p}{p} = 2\lambda\ln\frac{1/2+\lambda}{1/2-\lambda} = 2\lambda\ln(1 + \frac{2\lambda}{1/2-\lambda}) \leq \frac{4\lambda^2}{1/2-\lambda} = O(\lambda^2) = O((1-2p)^2)$, matching the query complexity lower bound within an $O(\log n)$ factor.*

**Algorithm. 1** The algorithm that we propose is completely deterministic and has several phases.

*Phase 1: Selecting a small subgraph.* Let $c = \frac{16}{(1-2p)^2}$.

1. Select $c \log n$ vertices arbitrarily from $V$. Let $V'$ be the set of selected vertices. Create a subgraph $G' = (V', E')$ by querying for every $(u, v) \in V' \times V'$ and assigning a weight of $\omega(u, v) = +1$ if the query answer is "yes" and $\omega(u, v) = -1$ otherwise .
2. Extract the heaviest weight subgraph $S$ in $G'$. If $|S| \geq c \log n$, move to Phase 2.
3. Else we have $|S| < c \log n$. Select a new vertex $u$, add it to $V'$, and query $u$ with every vertex in $V' \setminus \{u\}$. Move to step (2).

*Phase 2: Creating an Active List of Clusters.* Initialize an empty list called active when Phase 2 is executed for the first time.

1. Add $S$ to the list active.
2. Update $G'$ by removing $S$ from $V'$ and every edge incident on $S$. For every vertex $z \in V'$, if $\sum_{u \in S} \omega(z, u) > 0$, include $z$ in $S$ and remove $z$ from $G'$ with all edges incident to it.
3. Extract the heaviest weight subgraph $S$ in $G'$. If $|S| \geq c \log n$, Move to step(1). Else move to Phase 3.

*Phase 3: Growing the Active Clusters.* We now have a set of clusters in active.

1. Select an unassigned vertex $v$ not in $V'$ (that is previously unexplored), and for every cluster $\mathcal{C} \in$ active, pick $c \log n$ distinct vertices $u_1, u_2, ...., u_l$ in the cluster and query $v$ with them. If the majority of these answers are "yes", then include $v$ in $\mathcal{C}$.

2. Else we have for every $\mathcal{C} \in$ active the majority answer is "no" for $v$. Include $v \in V'$ and query $v$ with every node in $V' \setminus v$ and update $E'$ accordingly. Extract the heaviest weight subgraph $S$ from $G'$ and if its size is at least $c \log n$ move to Phase 2 step (1). Else move to Phase 3 step (1) by selecting another unexplored vertex.

*Phase 4: Maximum Likelihood (ML) Estimate.*

1. When there is no new vertex to query in Phase 3, extract the maximum likelihood clustering of $G'$ and return them along with the active clusters, where the ML estimation is defined as,

$$\max_{S_\ell, \ell=1,\cdots: V = \sqcup_{\ell=1} S_\ell} \sum_\ell \sum_{i,j \in S_\ell, i \neq j} \omega_{i,j}, \quad \text{(see Appendix B.1)} \tag{1}$$

**Analysis.** The high level steps of the analysis are as follows. Suppose all $\binom{n}{2}$ queries on $V \times V$ have been made. If the ML estimate of the clustering with these $\binom{n}{2}$ answers is same as the true clustering of $V$ that is, $\sqcup_{i=1}^k V_i \equiv \sqcup_{i=1}^k \hat{V}_i$ then the algorithm for noisy oracle finds the true clustering with high probability.

Let without loss of generality, $|\hat{V}_1| \geq ... \geq |\hat{V}_l| \geq 6c \log n > |\hat{V}_{l+1}| \geq ... \geq |\hat{V}_k|$. We will show that Phase 1-3 recover $\hat{V}_1, \hat{V}_2...\hat{V}_l$ with probability at least $1 - \frac{1}{n}$. The remaining clusters are recovered in Phase 4.

A subcluster is a subset of nodes in some cluster. Lemma 2 shows that any set $S$ that is included in active in Phase 2 of the algorithm is a subcluster of $V$. This establishes that all clusters in active at any time are subclusters of some original cluster in $V$.

**Lemma 2.** *Let $c' = 6c = \frac{96}{(1-2p)^2}$. Algorithm 1 in Phase 1 and 3 returns a subcluster of $V$ of size at least $c \log n$ with high probability if $G'$ contains a subcluster of $V$ of size at least $c' \log n$. Moreover, it does not return any set of vertices of size at least $c \log n$ if $G'$ does not contain a subcluster of $V$ of size at least $c \log n$.*

Lemma 2 is proven in three steps. Step 1 shows that if $V'$ contains a subcluster of size $\geq c' \log n$ then $S \subseteq V_i$ for some $i \in [1,k]$ will be returned with high probability when $G'$ is processed. Step 2 shows that size of $S$ will be at least $c \log n$, and finally step 3 shows that if there is no subcluster of size at least $c \log n$ in $V'$, then no subset of size $> c \log n$ will be returned by the algorithm when processing $G'$, because otherwise that $S$ will span more than one cluster, and the weight of a subcluster contained in $S$ will be higher than $S$ giving to a contradiction.

From Lemma 2, any $S$ added to active in Phase 2 is a subcluster with high probability, and has size at least $c \log n$. Moreover, whenever $G'$ contains a subcluster of $V$ of size at least $c' \log n$, it is retrieved by the algorithm and added to active. The next lemma shows that each subcluster added to active is correctly grown to the true cluster: (1) every vertex added to such a cluster is correct, and (2) no two clusters in active can be merged. Therefore, clusters obtained from active are the true clusters.

**Lemma 3.** *The list* active *contains all the true clusters of $V$ of size $\geq c' \log n$ at the end of the algorithm with high probability.*

Finally, once all the clusters in active are grown, we have a fully queried graph in $G'$ containing the small clusters which can be retrieved in Phase 4. This completes the correctness of the algorithm. With the following lemma, we get Theorem 2.

**Lemma 4.** *The query complexity of the algorithm for faulty oracle is $O\left(\frac{nk \log n}{(1-2p)^2}\right)$.*

Running time of this algorithm is dominated by finding the heaviest weight subgraph in $G'$, execution of each of those calls can be done in time $O([\frac{k \log n}{(2p-1)^2}]^{O(\frac{\log n}{(2p-1)^2})})$, that is quasi-polynomial in $n$. We show that it is unlikely that this running time can be improved by showing a reduction from the famous *planted clique problem* for which quasi-polynomial time is the best known (see Appendix B.1).

## 3.2 Computationally Efficient Algorithm

We now prove the following theorem. We give the algorithm here which is completely deterministic with known $k$. The extension to unknown $k$ and a detailed proof of correctness are deferred to Appendix B.2.

**Theorem 3.** *There exists a polynomial time algorithm with query complexity $O(\frac{nk^2}{(2p-1)^4})$ for* Query-Cluster *with error probability $p$, which recovers all clusters of size at least $\Omega(\frac{k \log n}{(2p-1)^4})$.*

**Algorithm 2.** Let $N = \frac{64k^2 \log n}{(1-2p)^4}$. We define two thresholds $T(a) = pa + \frac{6}{(1-2p)}\sqrt{N \log n}$ and $\theta(a) = 2p(1-p)a + 2\sqrt{N \log n}$. The algorithm is as follows.

*Phase 1-2C: Selecting a Small Subgraph.* Initially we have an empty graph $G' = (V', E')$, and all vertices in $V$ are unassigned to any cluster.

1. Select $X$ new vertices arbitrarily from the unassigned vertices in $V \setminus V'$ and add them to $V'$ such that the size of $V'$ is $N$. If there are not enough vertices left in $V \setminus V'$, select all of them. Update $G' = (V', E')$ by querying for every $(u, v)$ such that $u \in X$ and $v \in V'$ and assigning a weight of $\omega(u, v) = +1$ if the query answer is "yes" and $\omega(u, v) = -1$ otherwise .
2. Let $N^+(u)$ denote all the neighbors of $u$ in $G'$ connected by $+1$-weighted edges. We now cluster $G'$. Select every $u$ and $v$ such that $u \neq v$ and $|N^+(u)|, |N^+(v)| \geq T(|V'|)$. Then if $|N^+(u)\setminus N^+(v)| + |N^+(v)\setminus N^+(u)| \leq \theta(|V'|)$ (the symmetric difference of these neighborhoods) include $u$ and $v$ in the same cluster. Include in active all clusters formed in this step that have size at least $\frac{64k \log n}{(1-2p)^4}$. If there is no such cluster, abort. Remove all vertices in such cluster from $V'$ and any edge incident on them from $E'$.

*Phase 3C: Growing the Active Clusters.*
1. For every unassigned vertex $v \in V \setminus V'$, and for every cluster $\mathcal{C} \in$ active, pick $c \log n$ distinct vertices $u_1, u_2, ...., u_l$ in the cluster and query $v$ with them. If the majority of these answers are "yes", then include $v$ in $\mathcal{C}$.
2. Output all the clusters in active and move to Phase 1 step (1) to obtain the remaining clusters.

Running time of the algorithm can be shown to be $O(\frac{nk \log n}{(1-2p)^2} + kN^\omega)$ where $\omega \leq 2.373$ is the exponent of fast matrix multiplication[5]. Thus for small values of $k$, we get a highly efficient algorithm. The query complexity of the algorithm is $O(\frac{nk^2 \log n}{(2p-1)^4})$ since each vertex is involved in at most $O(\frac{k^2 \log n}{(2p-1)^4})$ queries within $G'$ and $O(\frac{k \log n}{(2p-1)^2})$ across the active clusters. In fact, in each iteration, the number of queries within $G'$ is $O(N^2)$ and since there could be at most $k$ rounds, the overall query complexity is $O(\frac{nk \log n}{(2p-1)^2} + \min(\frac{nk^2 \log n}{(2p-1)^4}, kN^2))$. Moreover, using the algorithm for unknown $k$ verbatim, we can obtain a correlation clustering algorithm for random noise model that recovers all clusters of size $\Omega(\frac{\min(k, \sqrt{n}) \log n}{(2p-1)^4})$, improving over [6, 44] for $k < \frac{\sqrt{n}}{\log n}$ since our ML estimate on $G'$ is correlation clustering.

### 3.3 Non-adaptive Algorithm

Finally for non-adaptive querying that is when querying must be done up front we prove the following. This shows while for $k = 2$, nonadaptive algorithms are as powerful as adaptive algorithms, for $k \geq 3$, substantial advantage can be gained by allowing adaptive querying. For details, see Appendix B.3 in the supplementary material.

**Theorem 4.** • *For $k = 2$, there exists an $O(n \log n)$ time nonadaptive algorithm that recovers the clusters with high probability with query complexity $O(\frac{n \log n}{(1-2p)^4})$. For $k \geq 3$, if $R$ is the ratio between the maximum to minimum cluster size, then there exists a randomized nonadaptive algorithm that recovers all clusters with high probability with query complexity $O(\frac{Rnk \log n}{(1-2p)^2})$. Moreover, there exists a computationally efficient algorithm for the same with query complexity $O(\frac{Rnk^2 \log n}{(1-2p)^4})$.*

• *For $k \geq 3$, if the minimum cluster size is $r$, then any deterministic nonadaptive algorithm must make $\Omega(\frac{n^2}{r})$ queries even when query answers are perfect to recover the clusters exactly. This shows that adaptive algorithms are much more powerful than their nonadaptive counterparts.*

## 4   Experiments

In this section, we report some experimental results on real and synthetic datasets.

**Real Datasets.**   We use the following three real datasets where the answers are generated from Amazon Mechanical Turk.

- `landmarks` consists of images of famous landmarks in Paris and Barcelona. Since the images are of different sides and clicked at different angles, it is difficult for humans to label them correctly. It consists of 266 nodes, 12 clusters with a total of 35245 edges, out of which 3738 are intra-cluster edges [31].
- `captcha` consists of CAPTCHA images, each showing a four-digit number. It consists of 244 nodes, 69 clusters with a total of 29890 edges out of which only 386 are intra-cluster edges [52].
- `gym` contains images of gymnastics athletes, where it is very difficult to distinguish the face of the athlete, e.g. when the athlete is upside down on the uneven bars. It consists of 94 nodes, 12 clusters and 4371 edges out of which 449 are intra-cluster edges [52].

**Repeating queries vs no repetition.**   Interestingly, we make the following observations. In `landmarks` dataset, when a majority vote is taken after asking each pairwise query 10 times, we get a total erroneous answers of 3696. However, just using the first crowd answer, the erroneous answers reduce to 2654. This shows that not only a simple strategy of repeating each query and taking a majority vote does not help to reduce error, in fact, it can amplify errors due to correlated answers by the crowd members. We observed the same phenomenon in the `gym` dataset where 449 answers are incorrect when majority voting is used over five answers for each query, compared to 310 by just using the first crowd user. For `captcha`, the error rate slightly decreases when using majority voting from 241 erroneous answers to 201.

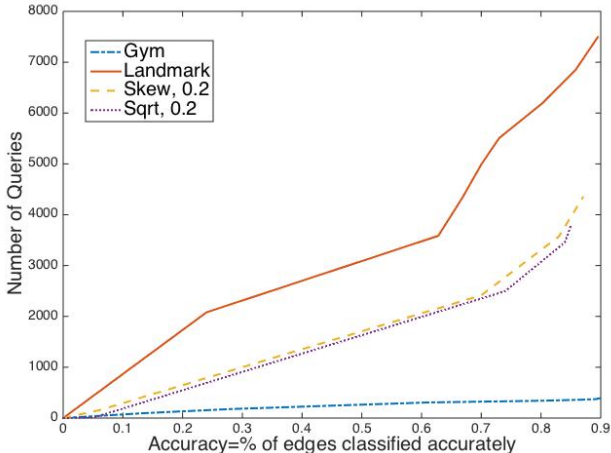

Figure 1: Number of Queries vs Accuracy Trade-off

**Synthetic Datasets.** We also did experiments on the following synthetic datasets from [27].

- `skew` and `sqrtn` contain fictitious hospital patients data, including name, phone number, birth date and address. The errors are generated synthetically with error probability $p = 0.2$. Each of them have 900 nodes, 404550 edges. `skew` has 8175 intra-cluster edges, whereas `sqrtn` contains 13050 intra-cluster edges.

**Number of Queries vs Accuracy.** Figure 1 plots the number of queries vs accuracy trade-off of our computationally efficient adaptive algorithm. Among the vertices that are currently clustered, we count the number of induced edges that are classified correctly and then divide it by the total number of edges in the dataset to calculate accuracy. Given the gap between maximum and minimum cluster size is significant in all real datasets, non-adaptive algorithms do not perform well. Moreover, if we select queries randomly, and look at the queried edges in each cluster, then even to achieve an intra-cluster minimum degree of two in every reasonable sized cluster, we waste a huge number queries on inter-cluster edges. While we make only 389 queries in gym to get an accuracy of 90%, the total number of random queries is 1957 considering only the clusters of size at least nine. For `landmark` dataset, the number of queries is about 7400 to get an accuracy of 90%, whereas the total number of random queries is 21675 considering the clusters of size at least seven. This can be easily explained by the huge discrepancy in the number of intra and inter-cluster edges where random edge querying cannot perform well. Among the edges that were mislabeled by our adaptive algorithm, $70 - 90\%$ of them are inter-cluster with very few errors in intra-cluster edges, that is the clusters returned are often superset of the original clusters. Similarly, the querying cost is also dominated by the inter-cluster edge queries. For example, out of 4339 queries issued by `skew`, 3844 are for inter-cluster edges. By using some side information such as a similarity matrix, a significant reduction in query complexity may be possible.

**Acknowledgements:** This work is supported in parts by NSF awards CCF 1642658, CCF 1642550, CCF 1464310, CCF 1652303, a Yahoo ACE Award and a Google Faculty Research Award. The authors are thankful to an anonymous reviewer whose comments led to many improvements in the presentation. The authors would also like to thank Sanjay Subramanian for his help with the experiments.

## Footnotes

[1]An approximation of $p$ can often be estimated manually from a small sample of crowd answers.

[2]Most recent works consider the region of interest as $p' = \frac{a \log n}{n}$ and $q' = \frac{b \log n}{n}$ for some $a > b > 0$.

[3] High probability implies with probability $1 - o_n(1)$, where $o_n(1) \to 0$ as $n \to \infty$

[4]For exact dependency on $p$ see the corresponding section.

[5]Fast matrix multiplication can be avoided by slightly increasing the dependency on $k$.

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
