[Supplementary Material · nips-supplementary.pdf]

# Clustering with Noisy Queries (Supplementary Material)

## Arya Mazumdar and Barna Saha

## A  Proof of Lemma 1

**Lemma** (Restating Lemma 1). *Suppose, there are $k$ clusters. There exist at least $\frac{4k}{5}$ clusters such that for each element $v$ from any of these clusters, $v$ will be assigned to a wrong cluster by any randomized algorithm with probability $0.29 - 10/k$ unless the total number of queries involving $v$ is more than $\frac{k}{10\Delta(p\|q)}$.*

*Proof.* Our first task is to cast the problem as a hypothesis testing problem.

**Step 1: Setting up the hypotheses.** Let us assume that the $k$ clusters are already formed, and we can moreover assume that all elements except for one element $v$ has already been assigned to a cluster. Note that, queries that do not involve the said element plays no role in this stage.

Now the problem reduces to a hypothesis testing problem where the $i$th hypothesis $H_i$ for $i = 1, \ldots, k$, denotes that the true cluster for $v$ is $V_i$. We can also add a null-hypothesis $H_0$ that stands for the vertex belonging to none of the clusters (since $k$ is unknown this is a hypothetical possibility for any algorithm[6]). Let $P_i$ denote the joint probability distribution of our observations (the answers to the queries involving vertex $v$) when $H_i$ is true, $i = 1, \ldots, k$. That is for any event $\mathcal{A}$ we have

$$P_i(\mathcal{A}) = \Pr(\mathcal{A}|H_i).$$

Suppose $T$ denotes the total number of queries made by a (possibly randomized) algorithm at this stage before assigning a cluster. Also let $\underline{x}$ be the $T$ dimensional binary vector that is the result of the queries. The assignment is based on $\underline{x}$. Let the random variable $T_i$ denote the number of queries involving cluster $V_i, i = 1, \ldots, k$. In the second phase, we need to identify a set of clusters that are not being queried with enough by the algorithm.

**Step 2: A set of "weak" clusters.** We must have, $\sum_{i=1}^k \mathbb{E}_0 T_i = T$. Let

$$J_1 \equiv \{i \in \{1, \ldots, k\} : \mathbb{E}_0 T_i \le \frac{10T}{k}\}.$$

Since, $(k - |J_1|)\frac{10T}{k} \le T$, we have $|J_1| \ge \frac{9k}{10}$. That is there exist at least $\frac{9k}{10}$ clusters in each of where less than $\frac{10T}{k}$ (on average under $H_0$) queries were made before assignment.

Let $\mathcal{E}_i \equiv \{$ the algorithm outputs cluster $V_i\}$. Let

$$J_2 = \{i \in \{1, \ldots, k\} : P_0(\mathcal{E}_i) \le \frac{10}{k}\}.$$

Moreover, since $\sum_{i=1}^k P_0(\mathcal{E}_i) \le 1$ we must have, $(k - |J_2|)\frac{10}{k} \le 1$, or $|J_2| \ge \frac{9k}{10}$. Therefore, $J = J_1 \cap J_2$ has size,

$$|J| \ge 2 \cdot \frac{9k}{10} - k = \frac{4k}{5}.$$

Now let us assume that we are given an element $v \in V_j$ for some $j \in J$ to cluster ($H_j$ is the true hypothesis). The probability of correct clustering is $P_j(\mathcal{E}_j)$. In the last step, we give an upper bound on probability of correct assignment for this element.

**Step 3: Bounding probability of correct assignment for weak cluster elements.** We must have,

$$P_j(\mathcal{E}_j) = P_0(\mathcal{E}_j) + P_j(\mathcal{E}_j) - P_0(\mathcal{E}_j)$$

$$\leq \frac{10}{k} + |P_0(\mathcal{E}_j) - P_j(\mathcal{E}_j)|$$

$$\leq \frac{10}{k} + \|P_0 - P_j\|_{TV} \leq \frac{10}{k} + \sqrt{\frac{1}{2}D(P_0\|P_j)}.$$

where we again used the definition of the total variation distance and in the last step we have used the Pinsker's inequality [20]. The task is now to bound the divergence $D(P_0\|P_j)$. Recall that $P_0$ and $P_j$ are the joint distributions of the independent random variables (answers to queries) that are identical to one of two Bernoulli random variables: $Y$, which is Bernoulli($p$), or $Z$, which is Bernoulli($q$). Let $X_1, \ldots, X_T$ denote the outputs of the queries, all independent random variables. We must have, from the chain rule [20],

$$D(P_0\|P_j) = \sum_{i=1}^{T} D(P_0(x_i|x_1,\ldots,x_{i-1})\|P_j(x_i|x_1,\ldots,x_{i-1}))$$

$$= \sum_{i=1}^{T} \sum_{(x_1,\ldots,x_{i-1})\in\{0,1\}^{i-1}} P_0(x_1,\ldots,x_{i-1})D(P_0(x_i|x_1,\ldots,x_{i-1})\|P_j(x_i|x_1,\ldots,x_{i-1})).$$

Note that, for the random variable $X_i$, the term $D(P_0(x_i|x_1,\ldots,x_{i-1})\|P_j(x_i|x_1,\ldots,x_{i-1}))$ will contribute to $D(q\|p)$ only when the query involves the cluster $V_j$. Otherwise the term will contribute to 0. Hence,

$$D(P_0\|P_j) = \sum_{i=1}^{T} \sum_{(x_1,\ldots,x_{i-1})\in\{0,1\}^{i-1}:i\text{th query involves }V_j} P_0(x_1,\ldots,x_{i-1})D(q\|p)$$

$$= D(q\|p)\sum_{i=1}^{T} \sum_{(x_1,\ldots,x_{i-1})\in\{0,1\}^{i-1}:i\text{th query involves }V_j} P_0(x_1,\ldots,x_{i-1})$$

$$= D(q\|p)\sum_{i=1}^{T} P_0(i\text{th query involves }V_j) = D(q\|p)\mathbb{E}_0 T_j \leq \frac{10T}{k}D(q\|p).$$

Now plugging this in,

$$P_j(\mathcal{E}_j) \leq \frac{10}{k} + \sqrt{\frac{1}{2}\frac{10T}{k}D(q\|p)} \leq \frac{10}{k} + \sqrt{\frac{1}{2}},$$

if $T \leq \frac{k}{10D(q\|p)}$. Had we bounded the total variation distance with $D(P_j\|P_0)$ in the Pinsker's inequality then we would have $D(p\|q)$ in the denominator. Obviously the smaller of $D(p\|q)$ and $D(q\|p)$ would give the stronger lower bound. $\qquad\square$

## B  Algorithms

In this section, we first develop an information theoretically optimal algorithm for clustering with faulty oracle within an $O(\log n)$ factor of the optimal query complexity. Next, we show how the ideas can be extended to make it computationally efficient. We consider both the adaptive and non-adaptive versions. All the missing proofs are presented here.

### B.1  Information-Theoretic Optimal Algorithm

We restate the algorithm.

Let $V = \sqcup_{i=1}^{k} V_i$ be the true clustering and $V = \sqcup_{i=1}^{k} \hat{V}_i$ be the maximum likelihood (ML) estimate of the clustering that can be found when all $\binom{n}{2}$ queries have been made to the faulty oracle. Our first result obtains a query complexity upper bound within an $O(\log n)$ factor of the information theoretic optimal algorithm. The algorithm runs in quasi-polynomial time, and we show this is the optimal possible assuming the famous *planted clique* hardness. In Section 3.2, we develop a computationally efficient algorithm for clustering with noisy oracle.

**Theorem** (restated 2). *There exists an algorithm with query complexity $O(\frac{nk \log n}{(1-2p)^2})$ for* Query-Cluster *that returns the ML estimate with high probability when query answers are incorrect with probability $p < \frac{1}{2}$. Moreover, the algorithm returns all true clusters of $V$ of size at least $\frac{C \log n}{(1-2p)^2}$ for a suitable constant $C$ with probability $1 - o_n(1)$.*

**Remark 2.** *Assuming $p = \frac{1}{2} - \lambda$, as $\lambda - > 0$, $\Delta(p\|1-p) = O(\lambda^2) = O((2p-1)^2)$. Thus our upper bound is within a $\log n$ factor of the information theoretic optimum in this range.*

**Algorithm. 1** The algorithm that leads us to the above theorem has several phases. The main idea is as follows. We start by selecting a small subset of vertices, and extract the heaviest weight subgraph in it by suitably defining edge weight. If the subgraph extracted has $\sim \log n$ size, we are confident that it is part of an original cluster. We then grow it completely, where a decision to add a new vertex to it happens by considering the query answers involving these different $\log n$ vertices and the new vertex. Otherwise, if the subgraph extracted has size less than $\log n$, we select more vertices. We note that we would never have to select more than $O(k \log n)$ vertices, because by pigeonhole principle, this will ensure that we have selected at least $\sim \log n$ members from a cluster, and the subgraph detected will have size at least $\log n$. This helps us to bound the query complexity. We note that our algorithm is completely deterministic.

*Phase 1: Selecting a small subgraph.* Let $c = \frac{16}{(1-2p)^2}$.
1. Select $c \log n$ vertices arbitrarily from $V$. Let $V'$ be the set of selected vertices. Create a subgraph $G' = (V', E')$ by querying for every $(u,v) \in V' \times V'$ and assigning a weight of $\omega(u,v) = +1$ if the query answer is "yes" and $\omega(u,v) = -1$ otherwise .
2. Extract the heaviest weight subgraph $S$ in $G'$. If $|S| \geq c \log n$, move to Phase 2.
3. Else we have $|S| < c \log n$. Select a new vertex $u$, add it to $V'$, and query $u$ with every vertex in $V' \setminus \{u\}$. Move to step (2).

*Phase 2: Creating an Active List of Clusters.* Initialize an empty list called active when Phase 2 is executed for the first time.
1. Add $S$ to the list active.
2. Update $G'$ by removing $S$ from $V'$ and every edge incident on $S$. For every vertex $z \in V'$, if $\sum_{u \in S} \omega(z, u) > 0$, include $z$ in $S$ and remove $z$ from $G'$ with all edges incident to it.
3. Extract the heaviest weight subgraph $S$ in $G'$. If $|S| \geq c \log n$, Move to step(1). Else move to Phase 3.

*Phase 3: Growing the Active Clusters.* We now have a set of clusters in active.
1. Select an unassigned vertex $v$ not in $V'$ (that is previously unexplored), and for every cluster $\mathcal{C} \in$ active, pick $c \log n$ distinct vertices $u_1, u_2, ...., u_l$ in the cluster and query $v$ with them. If the majority of these answers are "yes", then include $v$ in $\mathcal{C}$.
2. Else we have for every $\mathcal{C} \in$ active the majority answer is "no" for $v$. Include $v \in V'$ and query $v$ with every node in $V' \setminus v$ and update $E'$ accordingly. Extract the heaviest weight subgraph $S$ from $G'$ and if its size is at least $c \log n$ move to Phase 2 step (1). Else move to Phase 3 step (1) by selecting another unexplored vertex.

*Phase 4: Maximum Likelihood (ML) Estimate.*
1. When there is no new vertex to query in Phase 3, extract the maximum likelihood clustering of $G'$ and return them along with the active clusters, where the ML estimation is defined as,

$$\max_{S_\ell, \ell=1, \cdots : V = \sqcup_{\ell=1} S_\ell} \sum_{\ell} \sum_{i,j \in S_\ell, i \neq j} \omega_{i,j}, \quad \text{(see Lemma 8)} \qquad (2)$$

**Analysis.** To establish the correctness of the algorithm, we show the following. Suppose all $\binom{n}{2}$ queries on $V \times V$ have been made. If the ML estimate of the clustering with these $\binom{n}{2}$ answers is same as the true clustering of $V$ that is, $\sqcup_{i=1}^k V_i \equiv \sqcup_{i=1}^k \hat{V}_i$ then the algorithm for faulty oracle finds the true clustering with high probability.

Let without loss of generality, $|\hat{V}_1| \geq ... \geq |\hat{V}_l| \geq 6c \log n > |\hat{V}_{l+1}| \geq ... \geq |\hat{V}_k|$. We will show that Phase 1-3 recover $\hat{V}_1, \hat{V}_2 ... \hat{V}_l$ with probability at least $1 - \frac{1}{n}$. The remaining clusters are recovered in Phase 4.

A subcluster is a subset of nodes in some cluster. Lemma 2 shows that any set $S$ that is included in active in Phase 2 of the algorithm is a subcluster of $V$. This establishes that all clusters in active at any time are subclusters of some original cluster in $V$. Next, Lemma 3 shows that elements that are added to a cluster in active are added correctly, and no two clusters in active can be merged. Therefore, clusters obtained from active are the true clusters. Finally, the remaining of the clusters can be retrieved from $G'$ by computing a ML estimate on $G'$ in Phase 4, leading to Theorem 5.

We will use the following version of the Hoeffding's inequality heavily in our proof. We state it here for the sake of completeness.

Hoeffding's inequality for large deviation of sums of bounded independent random variables is well known [35][Thm. 2].

**Lemma 5** (Hoeffding). *If $X_1, \ldots, X_n$ are independent random variables and $a_i \leq X_i \leq b_i$ for all $i \in [n]$. Then*

$$\Pr(|\frac{1}{n}\sum_{i=1}^{n}(X_i - \mathbb{E}X_i)| \geq t) \leq 2\exp(-\frac{2n^2t^2}{\sum_{i=1}^{n}(b_i - a_i)^2}).$$

This inequality can be used when the random variables are independently sampled with replacement from a finite sample space. However due to a result in the same paper [35][Thm. 4], this inequality also holds when the random variables are sampled without replacement from a finite population.

**Lemma 6** (Hoeffding). *If $X_1, \ldots, X_n$ are random variables sampled without replacement from a finite set $\mathcal{X} \subset \mathbb{R}$, and $a \leq x \leq b$ for all $x \in \mathcal{X}$. Then*

$$\Pr(|\frac{1}{n}\sum_{i=1}^{n}(X_i - \mathbb{E}X_i)| \geq t) \leq 2\exp(-\frac{2nt^2}{(b-a)^2}).$$

**Lemma** (restated 2). *Let $c' = 6c = \frac{96}{(2p-1)^2}$. Algorithm 1 in Phase 1 and 3 returns a subcluster of $V$ of size at least $c\log n$ with high probability if $G'$ contains a subcluster of $V$ of size at least $c'\log n$. Moreover, it does not return any set of vertices of size at least $c\log n$ if $G'$ does not contain a subcluster of $V$ of size at least $c\log n$.*

*Proof.* Let $V' = \bigcup V_i'$, $i \in [1, k]$, $V_i' \cap V_j' = \emptyset$ for $i \neq j$, and $V_i' \subseteq V_i$. Suppose without loss of generality $|V_1'| \geq |V_2'| \geq \ldots \geq |V_k'|$. The lemma is proved via a series of claims.

**Claim 1.** *Let $|V_1'| \geq c'\log n$. Then a set $S \subseteq V_i$ for some $i \in [1, k]$ will be returned with high probability when $G'$ is processed.*

*Proof.* For an $i : |V_i'| \geq c'\log n$, we have

$$\mathbb{E}\sum_{s,t \in V_i', s<t} \omega_{s,t} = \binom{|V_i'|}{2}((1-p) - p) = (1-2p)\binom{|V_i'|}{2}.$$

Since $\omega_{s,t}$ are independent binary random variables, using the Hoeffding's inequality (Lemma 5),

$$\Pr\left(\sum_{s,t \in V_i', s<t} \omega_{s,t} \leq \mathbb{E}\sum_{s,t \in V_i', s<t} \omega_{s,t} - u\right) \leq e^{-\frac{u^2}{2\binom{|V_i'|}{2}}}.$$

Hence,

$$\Pr\left(\sum_{s,t \in V_i', s<t} \omega_{s,t} > (1-\delta)\mathbb{E}\sum_{s,t \in V_i', s<t} \omega_{s,t}\right) \geq 1 - e^{-\frac{\delta^2(1-2p)^2\binom{|V_i'|}{2}}{2}}.$$

Therefore with high probability (here the success probability is even $> 1 - \frac{1}{n^{\log n}}$)

$$\sum_{s,t \in V_i', s<t} \omega_{s,t} > (1-\delta)(1-2p)\binom{|V_i'|}{2}$$

$$\geq (1-\delta)(1-2p)\binom{c'\log n}{2} > \frac{c'^2}{3}(1-2p)\log^2 n,$$

for an appropriately chosen $\delta$ (say $\delta = \frac{1}{4}$).

So, when processing $G'$, the algorithm must return a set $S$ such that $|S| \geq c'\sqrt{\frac{2(1-2p)}{3}}\log n = c''\log n$ (define $c'' = c'\sqrt{\frac{2(1-2p)}{3}}$) with probability $> 1 - \frac{1}{n^{\log n}}$ - since otherwise

$$\sum_{i,j\in S, i<j} \omega_{i,j} < \binom{c'\sqrt{\frac{2(1-2p)}{3}}\log n}{2} < \frac{c'^2}{3}(1-2p)\log^2 n.$$

Now let $S \nsubseteq V_i$ for any $i$. Then $S$ must have intersection with at least 2 clusters. Let $V_i \cap S = C_i$ and let $j^* = \arg\min_{i:C_i\neq\emptyset}|C_i|$. We claim that,

$$\sum_{i,j\in S, i<j} \omega_{i,j} < \sum_{i,j\in S\setminus C_{j^*}, i<j} \omega_{i,j}, \tag{3}$$

with high probability. Condition (3) is equivalent to,

$$\sum_{i,j\in C_{j^*}, i<j} \omega_{i,j} + \sum_{i\in C_{j^*}, j\in S\setminus C_{j^*}} \omega_{i,j} < 0. \tag{I}$$

However this is true because,

1. $\mathbb{E}\left(\sum_{i,j\in C_{j^*}, i<j} \omega_{i,j}\right) = (1-2p)\binom{|C_{j^*}|}{2}$ and $\mathbb{E}\left(\sum_{i\in C_{j^*}, j\in S\setminus C_{j^*}} \omega_{i,j}\right) = -(1-2p)|C_{j^*}|\cdot|S\setminus C_{j^*}|$. Note that $|S\setminus C_{j^*}| \geq |C_{j^*}|$. Hence the expected value of the L.H.S. of (I) is negative.

2. As long as $|C_{j^*}| \geq \frac{12\sqrt{\log n}}{(1-2p)}$, we have from Hoeffding's inequality,

$$\Pr\left(\sum_{i,j\in C_{j^*}, i<j} \omega_{i,j} \geq (1+\nu)(1-2p)\binom{|C_{j^*}|}{2}\right)$$

$$\leq e^{-\frac{\nu^2(1-2p)^2\binom{|C_{j^*}|}{2}}{2}} = n^{-36\nu^2}.$$

While at the same time,

$$\Pr\left(\sum_{i\in C_{j^*}, j\in S\setminus C_{j^*}} \omega_{i,j} \geq -(1-\nu)(1-2p)|C_{j^*}|\cdot|S\setminus C_{j^*}|\right)$$

$$\leq e^{-\frac{\nu^2(1-2p)^2|C_{j^*}|\cdot|S\setminus C_{j^*}|}{2}} = n^{-72\nu^2}.$$

Setting $\nu = \frac{1}{4}$ (say), of course with high probability (probability at least $1 - \frac{2}{n^{2.25}}$)

$$\sum_{i,j\in C_{j^*}, i<j} \omega_{i,j} + \sum_{i\in C_{j^*}, j\in S\setminus C_{j^*}} \omega_{i,j} < 0.$$

3. When $|C_{j^*}| < \frac{12\sqrt{\log n}}{(1-2p)}$, let $|C_{j^*}| = x$. We have,

$$\sum_{i,j\in C_{j^*}, i<j} \omega_{i,j} \leq \binom{|C_{j^*}|}{2} \leq \frac{x^2}{2}.$$

While at the same time,

$$\Pr\left(\sum_{i\in C_{j^*}, j\in S\setminus C_{j^*}} \omega_{i,j} \geq -(1-\nu)(1-2p)|C_{j^*}|\cdot|S\setminus C_{j^*}|\right)$$

$$\leq e^{-\frac{\nu^2(1-2p)^2|C_{j^*}|\cdot|S\setminus C_{j^*}|}{2}} \leq e^{-\frac{\nu^2(1-2p)^2 x(|S|-x)}{2}}$$

If $x \geq \sqrt{\frac{3}{2(1-2p)}}$, then $x(|S| - x) \geq \frac{2x|S|}{3} = \frac{2c' \log n}{3} \geq \frac{64 \log n}{(1-2p)^2}$, where the second inequality followed since $x < \frac{S}{3}$. Hence, in this case, again setting $\nu = \frac{1}{4}$ and noting the value of $S$ and the fact $|C_{j^*}| < \frac{12\sqrt{\log n}}{(1-2p)}$, with probability at least $1 - \frac{1}{n^2}$,

$$\sum_{i,j \in C_{j^*}, i<j} \omega_{i,j} + \sum_{i \in C_{j^*}, j \in S \backslash C_{j^*}} \omega_{i,j} < 0.$$

If $x < \sqrt{\frac{3}{2(1-2p)}}$, then $(S - x) > \frac{48 x \log n}{(1-2p)}$. Hence $E[\sum_{i \in C_{j^*}, j \in S \backslash C_{j^*}} \omega_{i,j}] \leq -(1 - 2p)x(S - x) < -48 \log n \frac{x^2}{2}$.

Hence by Hoeffding's inequality,

$$\Pr \Big( \sum_{i \in C_{j^*}, j \in S \backslash C_{j^*}} \omega_{i,j} \geq -\frac{x^2}{2} \Big) \leq e^{-\frac{2*47*47 x^4 \log^2 n}{|C_{j^*}||S \backslash C_{j^*}|}} \leq e^{-\frac{2*47*47 x^3 \log^2 n}{|S|}} << \frac{1}{n^2}$$

Hence (3) is true with probability at least $1 - \frac{4}{n^2}$. But then the algorithm would not return $S$, but will return $S \backslash C_{j^*}$. Hence, we have run into a contradiction. This means $S \subseteq V_i$ for some $V_i$. $\qquad\square$

**Claim 2.** *Let $|V_1'| \geq c' \log n$. Then a set $S \subseteq V_i$ for some $i \in [1, k]$ with size at least $c \log n$ will be returned with high probability when $G'$ is processed.*

*Proof.* From Claim 1 with probability at least $1 - \frac{4}{n^2}$, $S \subseteq V_i$ and

$$\sum_{i,j \in S, i<j} \omega_{i,j} \geq \frac{c'^2}{3}(1 - 2p) \log^2 n.$$

Consider the situation that $|S| = x < c \log n = \frac{c' \log n}{6}$. Then

$$E[\sum_{i,j \in S, i<j} \omega_{i,j}] < \frac{x^2}{2}(1 - 2p)$$

Hence, by the Hoeffding's inequality

$$\Pr \Big( \sum_{i,j \in S, i<j} \omega_{i,j} \geq \frac{c'^2}{3}(1 - 2p) \log^2 n \Big) \leq e^{-\frac{(1-2p)^2 \left(\frac{c'^2}{3} \log^2 n - \frac{x^2}{2}\right)^2}{x^2}}$$

$$\leq e^{-\frac{(1-2p)^2 \left(\frac{c'^2}{4} \log^2 n\right)^2}{x^2}} << \frac{1}{n^2}$$

Therefore, $|S| \geq c \log n$ with probability at least $1 - \frac{5}{n^2}$.

$\qquad\square$

**Claim 3.** *If $|V_1'| < c \log n$. then no subset of size $> c \log n$ will be returned by the algorithm for faulty oracle when processing $G'$ with high probability.*

*Proof.* If the algorithm returns a set $S$ with $|S| > c \log n$ then $S$ must have intersection with at least 2 clusters in $V$. Now following the same argument as in Claim 1 to establish Eq. (3), we arrive to a contradiction, and $S$ cannot be returned. $\qquad\square$

Since, the algorithm attempts to extract a heaviest weight subgraph at most $n$ times, and each time the probability of failure is at most $O(\frac{1}{n^2})$. By union bound, all the calls succeed with probability at least $1 - O(\frac{1}{n})$. This establishes the lemma. $\qquad\square$

**Lemma** (restated 3). *The list* active *contains all the true clusters of $V$ of size $\geq c' \log n$ at the end of the algorithm with high probability.*

*Proof.* From Lemma 2, any cluster that is added to active in Phase 2 is a subset of some original cluster in $V$ with high probability, and has size at least $c \log n$. Moreover, whenever $G'$ contains a subcluster of $V$ of size at least $c' \log n$, it is retrieved by the algorithm and added to active.

When a vertex $v$ is added to a cluster $\mathcal{C}$ in active, we have $|\mathcal{C}| \geq c \log n$ at that time, and there exist $l = c \log n$ distinct members of $\mathcal{C}$, say, $u_1, u_2, .., u_l$ such that majority of the queries of $v$ with these vertices returned $+1$. Consider the situation that $v \notin \mathcal{C}$. Then the expected number of queries among the $l$ queries that had an answer "yes" ($+1$) is $lp$. We now use the following version of the Chernoff bound.

**Lemma 7** (Chernoff Bound). *Let $X_1, X_2, ..., X_n$ be independent binary random variables, and $X = \sum_{i=1}^{n} X_i$ with $E[X] = \mu$. Then for any $\epsilon > 0$*

$$\Pr[X \geq (1+\epsilon)\mu] \leq \exp\left(-\frac{\epsilon^2}{2+\epsilon}\mu\right)$$

*and,*

$$\Pr[X \leq (1-\epsilon)\mu] \leq \exp\left(-\frac{\epsilon^2}{2}\mu\right)$$

Hence, by the application of the Chernoff bound, $\Pr(v$ added to $\mathcal{C} \mid v \notin \mathcal{C}) \leq e^{-lp\frac{(\frac{1}{2p}-1)^2}{2+(\frac{1}{2p}-1)}} \leq \frac{1}{n^3}$.

On the other hand, if there exists a cluster $\mathcal{C} \in$ active such that $v \in \mathcal{C}$, then while growing $\mathcal{C}$, $v$ will be added to $\mathcal{C}$ (either $v$ already belongs to $G'$, or is a newly considered vertex). This again follows by the Chernoff bound. Here the expected number of queries to be answered "yes" is $(1-p)l$. Hence the probability that less than $\frac{l}{2}$ queries will be answered yes is $\Pr(v$ not included in $\mathcal{C} \mid v \in \mathcal{C}) \leq \exp(-c \log n(1-p)\frac{(1-2p)^2}{8(1-p)^2}) = \exp(-\frac{2}{(1-p)} \log n) \leq \frac{1}{n^2}$. Therefore, for all $v$, if $v$ is included in a cluster in active, the assignment is correct with probability at least $1 - \frac{1}{n}$. Also, the assignment happens as soon as such a cluster is formed in active and $v$ is explored (whichever happens first).

Furthermore, two clusters in active cannot be merged. Suppose, if possible there are two clusters $\mathcal{C}_1$ and $\mathcal{C}_2$ which ought to be subset of the same cluster in $V$. Let without loss of generality $\mathcal{C}_2$ is added later in active. Consider the first vertex $v \in \mathcal{C}_2$ that is considered by our algorithm. If $\mathcal{C}_1$ is already there in active at that time, then with high probability $v$ will be added to $\mathcal{C}_1$ in Phase 3. Therefore, $\mathcal{C}_1$ must have been added to active after $v$ has been considered by our algorithm and added to $G'$. Now, at the time $\mathcal{C}_1$ is added to $A$ in Phase 2, $v \in V'$, and again $v$ will be added to $\mathcal{C}_1$ with high probability in Phase 2–thereby giving a contradiction.

This completes the proof of the lemma. $\qquad\square$

**Theorem 5.** *If the ML estimate of the clustering of $V$ with all possible $\binom{n}{2}$ queries return the true clustering, then the algorithm for faulty oracle returns the true clusters with high probability. Moreover, it returns all the true clusters of $V$ of size at least $c' \log n$ with high probability.*

*Proof.* From Lemma 2 and Lemma 3, active contains all the true clusters of $V$ of size at least $c' \log n$ with high probability. Any vertex that is not included in the clusters in active at the end of the algorithm, are in $G'$. Also $G'$ contains all possible pairwise queries among them. Clearly, then the ML estimate of $G'$ will be the true ML estimate of the clustering restricted to these clusters. $\qquad\square$

**Lemma** (restated 4). *The query complexity of the algorithm for faulty oracle is $O\left(\frac{nk \log n}{(2p-1)^2}\right)$.*

*Proof.* Let there be $k'$ clusters in active when $v$ is considered by the algorithm. $k'$ could be 0 in which case $v$ is considered in Phase 1, else $v$ is considered in Phase 3. Therefore, $v$ is queried with at most $ck' \log n$ members, $c \log n$ each from the $k'$ active clusters. If $v$ is not included in one of these clusters, then $v$ is added to $G'$ and queried will all vertices $V'$ in $G'$. We have seen in the correctness proof (Lemma 5) that if $G'$ contains at least $c' \log n$ vertices from any original cluster, then ML estimate on $G'$ retrieves those vertices as a cluster with high probability. Hence, when $v$ is queried with the vertices in $G'$, $|V'| \leq (k - k')c' \log n$. Thus the total number of queries made when the algorithm considers $v$ is at most $c'k \log n$, where $c' = 6c = \frac{96}{(2p-1)^2}$ when the error probability is

$p$. This gives the query complexity of the algorithm considering all the vertices, which matches the lower bound computed in Section 2 within an $O(\log n)$ factor. □

Now combining all these we get the statement of Theorem 2.

**Running Time & Connection to Planted Clique** While the algorithm described above is very close to information theoretic optimal, the running time is not polynomial. Moreover, it is unlikely that the algorithm can be made efficient.

A crucial step of our algorithm is to find a large cluster of size at least $O(\frac{\log n}{(2p-1)^2})$, which can of course be computed in $O(n^{\frac{\log n}{(2p-1)^2}})$ time. However, since size of $G'$ is bounded by $O(\frac{k \log n}{(2p-1)^2})$, the running time to compute such a heaviest weight subgraph is $O([\frac{k \log n}{(2p-1)^2}]^{\frac{\log n}{(2p-1)^2}})$. This running time is unlikely to be improved to a polynomial. This follows from the planted clique conjecture.

**Conjecture 1** (Planted Clique Hardness). *Given an Erdős-Rényi random graph $G(n, p)$, with $p = \frac{1}{2}$, the planted clique conjecture states that if we plant in $G(n, p)$ a clique of size $t$ where $t = [\Omega(\log n), o(\sqrt{n})]$, then there exists no polynomial time algorithm to recover the largest clique in this planted model.*

**Reduction.** Given such a graph with a planted clique of size $t = \Theta(\log n)$, we can construct a new graph $H$ by randomly deleting each edge with probability $\frac{1}{3}$. Then in $H$, there is one cluster of size $t$ where edge error probability is $\frac{1}{3}$ and the remaining clusters are singleton with inter-cluster edge error probability being $\frac{1}{2} * \frac{2}{3} = \frac{1}{3}$. So, if we can detect the heaviest weight subgraph in polynomial time in the faulty oracle algorithm, then there will be a polynomial time algorithm for the planted clique problem.

In fact, the reduction shows that if it is computationally hard to detect a planted clique of size $t$ for some value of $t > 0$, then it is also computationally hard to detect a cluster of size $\leq t$ in the faulty oracle model. Note that $t = o(\sqrt{n})$. In the next section, we propose a computationally efficient algorithm which recovers all clusters of size at least $\frac{\min(k, \sqrt{n}) \log n}{(1-2p)^2}$ with high probability, which is the best possible assuming the conjecture, and can potentially recover much smaller sized clusters if $k = o(\sqrt{n})$.

**Finding the Maximum Likelihood Clustering of $V$ with faulty oracle** In proving Theorem 2 and Theorem 5, we used the fact that the ML estimate of $G'$ is given by Equation 2. We here give a proof.

We can view the clustering problem as following. We have an undirected graph $G(V \equiv [n], E)$, such that $G$ is a union of $k$ disjoint cliques $G_i(V_i, E_i)$, $i = 1, \ldots, k$. The subsets $V_i \in [n]$ are unknown to us; they are called the clusters of $V$. The adjacency matrix of $G$ is a block-diagonal matrix. Let us denote this matrix by $A = (a_{i,j})$.

Now suppose, each edge of $G$ is erased independently with probability $p$, and at the same time each non-edge is replaced with an edge with probability $p$. Let the resultant adjacency matrix of the modified graph be $Z = (z_{i,j})$. The aim is to recover $A$ from $Z$.

**Lemma 8.** *The maximum likelihood recovery is given by the following:*

$$\max_{S_\ell, \ell=1, \cdots : V = \sqcup_\ell S_\ell} \prod_\ell \prod_{i,j \in S_\ell, i \neq j} P_+(z_{i,j}) \prod_{r,t, r \neq t} \prod_{i \in S_r, j \in S_t} P_-(z_{i,j})$$

$$= \max_{S_\ell, \ell=1, \cdots : V = \sqcup_{\ell=1} S_\ell} \prod_\ell \prod_{i,j \in S_\ell, i \neq j} \frac{P_+(z_{i,j})}{P_-(z_{i,j})} \prod_{i,j \in V, i \neq j} P_-(z_{i,j}).$$

*where, $P_+(1) = 1 - p$, $P_+(0) = p$, $P_-(1) = p$, $P_-(0) = 1 - p$.*

Hence, the ML recovery asks for,

$$\max_{S_\ell, \ell=1, \cdots : V = \sqcup_{\ell=1} S_\ell} \sum_\ell \sum_{i,j \in S_\ell, i \neq j} \ln \frac{P_+(z_{i,j})}{P_-(z_{i,j})}.$$

Note that,

$$\ln \frac{P_+(0)}{P_-(0)} = -\ln \frac{P_+(1)}{P_-(1)} = \ln \frac{p}{1-p}.$$

Hence the ML estimation is,

$$\max_{S_\ell, \ell=1,\cdots: V=\sqcup_{\ell=1} S_\ell} \sum_\ell \sum_{i,j \in S_\ell, i \neq j} \omega_{i,j}, \tag{4}$$

where $\omega_{i,j} = 2z_{i,j} - 1, i \neq j$, i.e., $\omega_{i,j} = 1$, when $z_{i,j} = 1$ and $\omega_{i,j} = -1$ when $z_{i,j} = 0, i \neq j$. Further $\omega_{i,i} = z_{i,i} = 0, i = 1, \ldots, n$.

Note that (4) is equivalent to finding correlation clustering in $G$ with the objective of maximizing the consistency with the edge labels, that is we want to maximize the total number of positive intra-cluster edges and total number of negative inter-cluster edges [6, 44, 43]. This can be seen as follows.

$$\max_{S_\ell, \ell=1,\cdots: V=\sqcup_{\ell=1} S_\ell} \sum_\ell \sum_{i,j \in S_\ell, i \neq j} \omega_{i,j}$$

$$\equiv \max_{S_\ell, \ell=1,\cdots: V=\sqcup_{\ell=1} S_\ell} \Big[ \sum_\ell \sum_{i,j \in S_\ell, i \neq j} \big|(i,j) : \omega_{i,j} = +1\big| - \big|(i,j) : \omega_{i,j} = -1\big| \Big] + \sum_{i,j \in V, i \neq j} \big|(i,j) : \omega_{i,j} = -1\big|$$

$$= \max_{S_\ell, \ell=1,\cdots: V=\sqcup_{\ell=1} S_\ell} \Big[ \sum_\ell \sum_{i,j \in S_\ell, i \neq j} \big|(i,j) : \omega_{i,j} = +1\big| + \Big[ \sum_{r,t:r \neq t} \big|(i,j) : i \in S_r, j \in S_t, \omega_{i,j} = -1\big| \Big] \Big].$$

Therefore (4) is same as correlation clustering, however viewing it as obtaining clusters with maximum intra-cluster weight helps us to obtain the desired running time of our algorithm. Also, note that, we have a random instance of correlation clustering here, and not a worst case instance.

## B.2 Computationally Efficient Algorithm

**Known $k$** We first design an algorithm when $k$, the number of clusters is known. Then we extend it to the case of unknown $k$. The algorithm is completely deterministic.

**Theorem 6.** *There exists a polynomial time algorithm with query complexity $O(\frac{nk^2}{(2p-1)^4})$ for* Query-Cluster *with error probability $p$, which recovers all clusters of size at least $\Omega(\frac{k \log n}{(2p-1)^4})$.*

**Algorithm 2.** Let $N = \frac{64k^2 \log n}{(1-2p)^4}$. We define two thresholds $T(a) = pa + \frac{6}{(1-2p)}\sqrt{N \log n}$ and $\theta(a) = 2p(1-p)a + 2\sqrt{N \log n}$. The algorithm is as follows.

*Phase 1-2C: Select a Small Subgraph.* Initially we have an empty graph $G' = (V', E')$, and all vertices in $V$ are unassigned to any cluster.

1. Select $X$ new vertices arbitrarily from the unassigned vertices in $V \setminus V'$ and add them to $V'$ such that the size of $V'$ is $N$. If there are not enough vertices left in $V \setminus V'$, select all of them in $X$. Update $G' = (V', E')$ by querying for every $(u,v)$ such that $u \in X$ and $v \in V'$ and assigning a weight of $\omega(u,v) = +1$ if the query answer is "yes" and $\omega(u,v) = -1$ otherwise .
2. Let $N^+(u)$ denote all the neighbors of $u$ in $G'$ connected by $+1$-weighted edges. We now cluster $G'$. Select every $u$ and $v$ such that $u \neq v$ and $|N^+(u)|, |N^+(v)| \geq T(|V'|)$. Then if $|N^+(u) \setminus N^+(v)| + |N^+(v) \setminus N^+(u)| \leq \theta(|V'|)$ (the symmetric difference of these neighborhoods) include $u$ and $v$ in the same cluster. Include in active all clusters formed in this step that have size at least $\frac{64k \log n}{(1-2p)^4}$. If there is no such cluster, abort. Remove all vertices in such cluster from $V'$ and any edge incident on them from $E'$.

*Phase 3C: Growing the Active Clusters.*

1. For every unassigned vertex $v \in V \setminus V'$, and for every cluster $\mathcal{C} \in$ active, pick $\frac{16 \log n}{(1-2p)^2}$ distinct vertices, $u_1, u_2, \ldots, u_l$ in the cluster and query $v$ with them. If the majority of these answers are "yes", then include $v$ in $\mathcal{C}$.
2. Output all the clusters in active and move to Phase 1 step (1) to obtain the remaining clusters.

**Analysis.** Note that at every iteration, we consider a set of $X$ new vertices from $V \setminus V'$ which have not been previously included in any cluster considered in active, and query all pairs in $X \times V' \setminus V$. Let $A$ denote the fixed $n \times n$ matrix, where if $(i, j), i, j \in V$ is queried by the algorithm in any iteration, we include the query result there ($+1$ or $-1$), else the entry is empty which indicates that the pair was not queried by the entire run of the algorithm. This matrix $A$ has the property that for any entry $(i, j)$, if $i$ and $j$ belong to the same cluster and queried then $A(i, j) = +1$ with probability $(1 - p)$ and $A(i, j) = -1$ with probability $p$. On the other hand, if $i$ and $j$ belong to different clusters and queried then $A(i, j) = -1$ with probability $(1 - p)$ and $A(i, j) = +1$ with probability $p$. Note that the adjacency matrix of $G'$ in any iteration is a submatrix of $A$ which has no empty entry.

We first look at *Phase 1-2C*. At every iteration, our algorithm selects a submatrix of $A$ corresponding to $V' \times V'$ after step 1. This submatrix of $A$ has no empty entry. Let us call it $A'$. We show that if $V'$ contains any subcluster of size $\geq \frac{64k \log n}{(2p-1)^4}$, it is retrieved by step 2 with probability at least $1 - \frac{1}{n^2}$. In that case, the iteration succeeds. Now the submatrices from one iteration to the other iteration can overlap, so we can only apply union bound to obtain the overall success probability, but that suffices. The probability that in step 2, the algorithm fails to retrieve any cluster of size at least $\frac{64k \log n}{(2p-1)^4}$ in any iteration is at most $\frac{1}{n^2}$. The number of iterations is at most $k < n$, since in every iteration except possibly for the last one, $V'$ contains at least one subcluster of that size by a simple pigeonhole principle. This is because in every iteration except possibly for the last one $|V'| = \frac{64k^2 \log n}{(2p-1)^4}$, and there are at most $k$ clusters. Therefore, the probability that there exists at least one iteration which fails to retrieve the "large" clusters is at most $\frac{k}{n^2} \leq \frac{1}{n}$ by union bound. Thus all the iterations will be successful in retrieving the large clusters with probability at least $1 - \frac{1}{n}$.

Now, following the same argument as Lemma 3, each such cluster will be grown completely by Phase 3-C step (1), and will be output correctly in Phase 3-C step 2.

**Lemma 9.** *Let $c = \frac{64}{(1-2p)^4}$. Whenever $G'$ contains a subcluster of size $ck \log n$, it is retrieved by Algorithm 2 in Phase 1-2C with high probability.*

*Proof.* Consider a particular iteration. Let $N^+(u)$ denote all the neighbors of $u$ in $G'$ connected by $+1$ edges. Let $A'$ denote the corresponding submatrix of $A$ corresponding to $G'$. We have $|V'| \leq N$ ($|V'| = N$ except possibly for the last iteration). Assume, $|V'| = N'$. Also $|V| = n$.

Let $C_u$ denote the cluster containing $u$. We have

$$E[|N^+(u)|] = (1 - p)|C_u| + p(N' - |C_u|) = pN' + (1 - 2p)|C_u|$$

By the Hoeffding's inequality

$$\Pr(|N^+(u)| \in pN' + (1 - 2p)|C_u| \pm 2\sqrt{N \log n}) \geq 1 - \frac{1}{n^4}$$

Therefore for all $u$ such that $|C_u| \geq \frac{8\sqrt{N \log n}}{(1-2p)^2}$, we have $|N^+(u)| > pN' + \frac{6}{(1-2p)}\sqrt{N \log n} = T(|V'|)$, and for all $u$ such that $|C_u| \leq \frac{4\sqrt{N \log n}}{(1-2p)^2}$, we have $|N^+(u)| < pN' + \frac{6}{(1-2p)}\sqrt{N \log n}$ with probability at least $1 - \frac{1}{n^3}$ by union bound.

Consider all $u$ such that $|N^+(u)| > T(|V'|)$. Then with probability at least $1 - \frac{1}{n^3}$, we have $|C_u| > \frac{4\sqrt{N \log n}}{(1-2p)^2}$. Let us call this set $U$. For every $u, v \in U, u \neq v$, the algorithm computes the symmetric difference of $N^+(u)$ and $N^+(v)$ which is

1. $2p(1 - p)N'$ on expectation if $u$ and $v$ belong to the same cluster. And again applying Hoeffding's inequality, it is at most $2p(1 - p)N' + 2\sqrt{N \log n}$ with probability at least $1 - \frac{1}{n^4}$.

2. $(p^2 + (1 - p)^2)(|C_u| + |C_v|) + 2p(1 - p)(N' - |C_u| - |C_v|) = 2p(1 - p)N' + (1 - 2p)^2(|C_u| + |C_v|)$ on expectation if $u$ and $v$ belong to different clusters. Again using the Hoeffding's inequality, it is at least $2p(1 - p)N' + (1 - 2p)^2(|C_u| + |C_v|) - 2\sqrt{N \log n}$ with probability at least $1 - \frac{1}{n^4}$.

Therefore, for all $u$ and $v$, either of the above two inequalities fail with probability at most $\frac{1}{n^2}$.

Now, since for all $u$ if $|N^+(u)| > T(|V'|)$ then $|C_u| > \frac{4\sqrt{N \log n}}{(1-2p)^2}$ with probability $1 - \frac{1}{n^3}$, we get

for every $u$ and $v$ in $U$, if the symmetric difference of $N^+(u)$ and $N^+(v)$ is $\leq 2p(1-p)N' + 2\sqrt{N \log n} = \theta(|V'|)$, then $u$ and $v$ must belong to the same cluster with probability at least $1 - \frac{1}{n^2} - \frac{1}{n^3} \geq 1 - \frac{2}{n^2}$.

Hence, all subclusters of $G'$ that have size at least $\frac{8\sqrt{N \log n}}{(1-2p)^2}$ will be retrieved correctly with probability at least $1 - \frac{2}{n^2}$. Now since $N' = N = \frac{64k^2 \log n}{(1-2p)^4}$ for all but possibly the last iteration, we have $\frac{8\sqrt{N \log n}}{(1-2p)^2} = \frac{64k \log n}{(1-2p)^4}$. Moreover, since there are at most $k$ clusters in $G$ and hence in $G'$, there exists at least one subcluster of size $\frac{64k \log n}{(1-2p)^4}$ in $G'$ in every iteration except possibly the last one, which will be retrieved.

Then, there could be at most $k < n$ iterations. The probability that in one iteration, the algorithm will fail to retrieve a large cluster by our analysis is at most $\frac{2}{n^2}$. Hence, by union bound over the iterations, the algorithm will successfully retrieve all clusters in Phase 1-2C with probability at least $1 - \frac{2}{n}$. $\square$

Now, following the same argument as in Lemma 3, each subcluster of size $\frac{64k \log n}{(1-2p)^4}$ will be grown completely by Phase 3-C step (1).

Running time of the algorithm is dominated by the time required to run step 2 of Phase 1-2C. Computing trivially, finding the symmetric differences of $+1$ neighborhoods all $\binom{N}{2}$ pairs requires time $O(N^3)$. We can keep a sorted list of $+1$ neighbors of every vertex is $O(N^2 \log n)$ time. Then, for every pair, it takes $O(N)$ time to find the symmetric difference. This can be reduced to $O(N^\omega)$ using fast matrix multiplication to compute set intersection where $\omega \leq 2.373$. Moreover, since each invocation of this step removes one cluster, there can be at most $k$ calls to it and for every vertex, time required in Phase 3C over all the rounds is $O(\frac{k \log n}{(1-2p)^2})$. This gives an overall running time of $O(\frac{nk \log n}{(1-2p)^2} + kN^\omega) = O(\frac{nk \log n}{(1-2p)^2} + k^{1+2\omega}) = O(\frac{nk \log n}{(1-2p)^2} + k^{5.746})$. Without fast matrix multiplication, the running time is $O(\frac{nk \log n}{(1-2p)^2} + k^7)$.

The query complexity of the algorithm is $O(\frac{nk^2 \log n}{(2p-1)^4})$ since each vertex is involved in at most $O(\frac{k^2 \log n}{(2p-1)^4})$ queries within $G'$ and $O(\frac{k \log n}{(2p-1)^2})$ across the active clusters. In fact, in each iteration, the number of queries within $G'$ is $O(N^2)$ and since there could be at most $k$ rounds, the overall query complexity is $O(\frac{nk \log n}{(2p-1)^2} + \min(\frac{nk^2 \log n}{(2p-1)^4}, kN^2))$. Thus we get Theorem 6. $\square$

**Remark 3.** *Readers familiar with the correlation clustering algorithm for noisy input from [6] would recognize that the idea of looking into symmetric difference of positive neighborhoods is from [6]. Like [6], we need to know the parameter $p$ to design our algorithm. In fact, one can view our algorithm as running the algorithm of [6] on carefully crafted subgraphs. Developing a parameter free algorithm that works without knowing $p$ remains an exciting future direction.*

**Unknown $k$**  Let $c = \frac{64}{(1-2p)^4}$. When the number of clusters $k$ is unknown, it is not possible exactly to determine when the subgraph $G' = (V', E')$ contains $ck^2 \log n$ sampled vertices. To overcome such difficulty, we propose the following approach of iteratively guessing and updating the estimate of $k$ based on the highest size of $N^+(v)$ for $v \in V'$. Let $\ell$ be the guessed value of $k$. We start with $\ell = 2$.

1. Randomly sample $X$ vertices so that $N = |V'| = c\ell^2 \log n$

2. For each $v \in V'$, estimate $\hat{C}_v = \frac{1}{(1-2p)}(|N^+(v)| - pN)$

3. If $\max_v \hat{C}_v > \frac{6\ell \log n}{(1-2p)^4}$ then run step 2 of Phase 1-3C on $G'$ with $k = \ell$, and then move to Phase 3C.

4. Else set $\ell = 2\ell$ and move to step (2).

Clearly, we will never guess $\ell > 2k$, and hence the process converges after at most $\log k$ rounds. When $N = c\ell^2 \log n$, we have $\sqrt{N \log n} \leq c\ell \log n$ (we must have $\ell^2 \leq n$, otherwise we sample the entire graph). From Lemma 9 we get, whenever $\hat{C}_v > \frac{6\ell \log n}{(1-2p)^4}$, the actual size of cluster containing $v$ is $\geq \frac{4\ell \log n}{(1-2p)^4}$ with high probability. We can then obtain the exact subcluster containing $v$ in $G'$ and grow it fully in Phase 3C with high probability. The query complexity remain the same within a factor of 2 and running time increases only by a factor of $\log k$.

**Discussion:** *Correlation Clustering over Noisy Input.* In a random noise model, also introduced by [6] and studied further by [44], we start with a ground truth clustering, and then each edge label is flipped with probability $p$. [6] gave an algorithm that recovers all true clusters of size $\geq c_1 \sqrt{n \log n}$ for some suitable constant $c_1$ under this model. Moreover, if all the clusters have size $\geq c_2 \sqrt{n}$, [44] gave a semi-definite programming based algorithm to recover all of them. Using the algorithm for unknown $k$ verbatim, we can obtain a correlation clustering algorithm for random noise model that recovers all clusters of size $\Omega(\frac{\min(k, \sqrt{n}) \log n}{(2p-1)^4})$. Since the maximum likelihood estimate of our algorithm is correlation clustering, the true clusters (which is same as the ML clustering) of size $\Omega(\frac{\min(k, \sqrt{n}) \log n}{(2p-1)^4})$ that the algorithm recovers is the correct correlation clustering output. Therefore, when $k < \frac{\sqrt{n}}{\log n}$, we can recover much smaller sized clusters than [6, 44].

**Theorem 7.** *There exists a deterministic polynomial time algorithm for correlation clustering over noisy input that recovers all the underlying true clusters of size at least $c_3 \min(k, \sqrt{n}) \log n$ for a suitable constant $c_3$ with high probability.*

### B.3 Non-adaptive Algorithm

In this section, we consider the case when all queries must be made upfront that is adaptive querying is not allowed. We show how our adaptive algorithms can be modified to handle such setting. Specifically, for $k = 2$, we show nonadaptive algorithms are as powerful as adaptive algorithms, but for $k \geq 3$, unless the maximum to minimum cluster size is bounded, there is a significant advantage gained by using adaptive algorithm.

We prove the following theorem.

**Theorem** (restated 4). • *For $k = 2$, there exists an $O(n \log n)$ time nonadaptive algorithm that recovers the clusters with high probability with query complexity $O(\frac{n \log n}{(1-2p)^4})$.*

• *For $k \geq 3$, if $R$ is the ratio between maximum to minimum cluster size, then there exists a randomized nonadaptive algorithm that recovers all clusters with high probability with query complexity $O(\frac{Rnk \log n}{(1-2p)^2})$. Moreover, there exists a computationally efficient algorithm for the same with query complexity $O(\frac{Rnk^2 \log n}{(1-2p)^4})$.*

• *For $k \geq 3$, if the minimum cluster size is $r$, then any deterministic non-adaptive algorithm must make $\Omega(\frac{n^2}{r})$ queries even when query answers are perfect to recover the clusters exactly. This shows that adaptive algorithms are much more powerful than their nonadaptive counterparts.*

**Non-adaptive with $k = 2$:** For $k = 2$, the algorithm is as follows. It constructs the graph $G' = (V', E')$ by randomly sampling $N = 4c \log n$ vertices where $c = \Theta(\frac{1}{(1-2p)^4})$ and querying all $\binom{|V'|}{2}$ pairs as well as all $(u, v)$ where $u \in V \setminus V'$ and $v \in V'$. Note that this is quite different from random querying.

$G'$ then contains at least one subcluster of size at least $2c \log n = \frac{N}{2}$, which is recovered by running the computationally efficient algorithm from Section 3.2. Using the query answers of $(u, v)$ where $u \in V \setminus V'$ and $v \in V'$, the subcluster is then grown fully. Finally, all the other vertices are put in a separate cluster.

The algorithm running time is $O(n \log n)$ from the running time discussion of our computationally efficient adaptive algorithm for known $k$. This improves upon [48, 16, 14].

**Non-adaptive with $k \geq 3$:** Let $R \geq 1$ be the ratio of the maximum to minimum cluster size. When the minimum size cluster is small, in Appendix C, we provide a lower bound of $\Omega(n^2)$ for

any deterministic algorithm. Our algorithm simply creates $G'$ by randomly and uniformly sampling $\Theta(\frac{Rk^2 \log n}{(1-2p)^4})$ vertices from $G$. It then queries all $(u,v) \in V' \times V'$. We here assume $\Theta(\frac{Rk^2 \log n}{(1-2p)^4}) < n$, otherwise $G'$ is the entire fully-queried graph $G$. The query complexity is therefore, $O(\frac{Rnk^2 \log n}{(1-2p)^4})$.

Since, we sample the vertices uniformly at random, the minimum number of vertices selected from any cluster with high probability using the Chernoff bound is $O(\frac{Rnk \log n}{(1-2p)^4})$. Now, again following the algorithm of Section 3.2, we can recover all these subclusters exactly with high probability–the remaining queries are then used to grow them fully. The running time of the algorithm is same as the running time of its adaptive version.

To obtain an information theoretic optimal result within an $O(\log n)$ factor, instead of sampling $\Theta(\frac{Rk^2 \log n}{(1-2p)^4})$ vertices, we sample $\Theta(\frac{Rk \log n}{(1-2p)^2})$ vertices from $G$ to construct $G'$ and then issue all pairwise queries $(u,v) \in V \times V'$. Then, by the same argument, the minimum size of any subcluster in $G'$ is at least $\Theta(\frac{\log n}{(1-2p)^2})$ with high probability which can be recovered by using the algorithm for detecting heaviest weight subgraph from Section 3.1.

## C  Lower Bound: Nonadaptive queries and the Stochastic Block Model

First, let us note that when there are only two clusters, and the oracle gives correct answers, then it is possible to recover the clusters with only $n-1$ queries. Indeed, just query every element with a fixed element. It is also easy to see than $\Omega(n)$ queries are required (since our lower bound of Theorem 1 is valid in this special case).

On the other hand, consider the case when there are $k > 2$ clusters, and the oracle is perfect. We show that any deterministic algorithm would require $\Omega(n^2)$ queries. This is in stark contrast with our adaptive algorithms which are all deterministic and achieve significantly less query complexity.

**Claim 4.** *Assume there are $k \geq 3$ clusters and the minimum size of a cluster is $r$. Then any deterministic nonadaptive algorithm must make $\Omega(\frac{n^2}{r})$ queries, even when the oracle is perfect.*

*Proof.* Consider a graph with $n$ vertices and there will be an edge between two vertices if the deterministic nonadaptive algorithm makes queries between them. Assume the number of queries made is at most $\frac{n^2}{4r}$. Then, using Turán's theorem, this graph must have an independent set of size at least $\frac{n}{n/2r+1} \approx 2r$. We can create an closeting instance with three clusters: one large cluster with $n - 2r$ vertices, and two small clusters with size $r$ each, where the union of the later two constitutes the independent set. Since the algorithm makes no query within the later two cluster, there will be no way to identify them. Hence the number of queries for any nonadaptive deterministic algorithm must be more than $\frac{n^2}{4r}$. $\qquad\square$

### C.1  Stochastic Block Model

Our model of faulty oracle is closely related to the stochastic block model. Indeed, if all $\binom{n}{2}$ queries are performed with the faulty oracle $\mathcal{O}_{p,q}$, we exactly recover the adjacency matrix of usual stochastic block model. When we are performing a fixed number $Q < \binom{n}{2}$ of queries to the oracle, we can think of that as a generalization of the stochastic block model, where only $Q$ entries of the adjacency matrix of the stochastic block model is being provided to us. One crucial point about our model is that though, we can adaptively query to carefully select the entries of the adjacency matrix of the stochastic block model to ensure recovery of the clustering.

Let us, consider the case when all of the $Q$ queries are made nonadaptively. This is still a generalization of stochastic block model (in which case $Q = \binom{n}{2}$). Assume the prior probability of each element being assigned to any cluster is uniform. Since each query involves two elements, this means that the average number of queries an element is involved in is $\frac{2Q}{n}$. Using Markov inequality, we can say that there exists at least $\frac{n}{2}$ elements $U$, each of which are involved in at most $\frac{4Q}{n}$ queries.

Now we can restrict ourselves to finding the clustering among only such $\frac{n}{2}$ elements each of which are involved in at most $\frac{4Q}{n}$ queries. Now let us just take any two clusters $V_1$ and $V_2$ and a fixed element $v \in V_1 \cap U$. We obtain $K = \frac{n}{2k}$ different equiprobable clusterings by interchanging $v$ with

the elements of $V_2 \cap U$. Let us consider the task of distinguishing between these $K$ hypotheses, by looking the query answers.

Now, we can use a generalized Fano's inequality from [50][Thm. 4], where we consider Renyi divergence of order $\frac{1}{2}$, to have,

$$-2\log\Big(\sqrt{\frac{1-P_e}{K}} + \sqrt{P_e(1-\frac{1}{K})}\Big)$$

$$\leq -\log\sum_y\big(\frac{1}{K}\sum_{j=1}^K\sqrt{Q_j(y)}\big)^2$$

where $P_e$ the probability of error of this hypothesis testing problem. This implies,

$$\Big(\sqrt{\frac{1-P_e}{K}} + \sqrt{P_e(1-\frac{1}{K})}\Big)^2 \geq 1 - \mathcal{H}^2(Q_i\|Q_j)$$

$$\geq 1 - \Big(1 - (1-\mathcal{H}^2(p\|q))^{\frac{8Q}{nk}}\Big) = (1-\mathcal{H}^2(p\|q))^{\frac{8Q}{nk}},$$

where we have used the fact that each element considered can influence at most $\frac{4Q}{nk}$ query answers on average by this interchange. Again, if we assume $p \sim \text{Bernoulli}\Big(\frac{a\log n}{n}\Big)$ and $q \sim \text{Bernoulli}\Big(\frac{b\log n}{n}\Big)$, a particular regime of interest for stochastic block model, then,

$$\sqrt{\frac{k}{n}} + \sqrt{P_e}$$

$$\geq \Big(\frac{\sqrt{ab}\log n}{n} + \sqrt{(1-\frac{a\log n}{n})(1-\frac{b\log n}{n})}\Big)^{\frac{4Q}{nk}}$$

$$= n^{-\Big(\frac{a+b}{2}-\sqrt{ab}-\frac{ab\log n}{n}\Big)\frac{4Q}{n^2k}}.$$

This implies, $\sqrt{P_e} \geq n^{-\Big(\frac{a+b}{2}-\sqrt{ab}\Big)\frac{4Q}{n^2k}} - \sqrt{k}n^{-1/2}$. In particular, if $\Big(\frac{a+b}{2}-\sqrt{ab}\Big)\frac{4Q}{n^2k} < \frac{1}{2}$, then $P_e > 0$. Hence, $P_e > \frac{1}{n}$ if $\sqrt{a}-\sqrt{b} < \frac{n}{2}\sqrt{\frac{k}{Q}}$.

Note that when $Q = \binom{n}{2}$, the maximum possible value, we get $\sqrt{a}-\sqrt{b} < \sqrt{\frac{k}{2}} \implies P_e > 0$,–this is slightly suboptimal by a factor of $\sqrt{2}$ than what is known for the stochastic block model [2, 49]. Tightening the constant, and getting matching upper bound for arbitrary $Q$ are interesting future work. However, note that, our tools are not specialized for this regime of stochastic block models, and the result works for general values of $Q$, not only the corner point of $Q = \binom{n}{2}$.

Now to extend this argument, to the case where adaptive querying is allowed, is difficult. Therefore we have to rely on the general technique of Theorem 1.

**Remark 4.** *There is another different version of Fano's inequality that we can use here - form [33][Thm. 7], that says the probability of error of this hypothesis testing problem is:*

$$P_e \geq 1 - \frac{\frac{4Q}{nk}(D(p\|q)+D(q\|p))+\ln 2}{\log\frac{n}{2k}}.$$

*This says that the number of nonadaptive queries must be at least $\Omega(\frac{nk\log n}{D(p\|q)+D(q\|p)})$ to recover the clustering with positive probability (this is indeed a lower bound for balanced clustering). As we have seen from Section 3.3, this bound is tight.*

## Footnotes

[6]This lower bound easily extends to the case even when $k$ is known.