[Reviews · NeurIPS 2017]

Reviewer 1



The noisy query model considered in this paper is as follows. One is allowed to ask queries of the form : do u,v belong to the same cluster? The oracle outputs +1 if they are and -1 otherwise, but it is allowed to make an error with a certain probability. Suppose the set of points to be clustered is denoted by V = V_1 \sqcup ... \sqcup V_k, with k clusters. The goal is minimize the number of queries and recover the exact clustering with a big enough probability. This paper gives a lower bound for the number of queries, also gives a quasi-polynomial time algorithm that achieves the lower bound upto a log(n) factor and finally gives computationally efficient algorithm with a worse query complexity. The paper claims to introduce the noisy query model for clustering. I am not aware of the literature, but this seems to be a very natural model to study and capture a lot of interesting real-life problems. The paper also has many interesting results which improves on previous papers in various regimes. In the symmetric case (that is, when the error probability for pairs belong to same cluster is same as the error probability for pairs belonging to different clusters), the paper gives a lower bound of Omega(nk/(1-2p)^2), and an algorithm that runs in O(n log n + k^6) time and makes at most O(nk^2 log n/(1-2p)^4). The experiments show a difference in the number of nonadaptive queries and adaptive queries required. Questions to the authors: 1) Could you elaborate on the experiments? What is the meaning of the sentence on line 363-364? Suppose you want to query if u,v belong to the same cluster, how many users do you ask this question and how do you determine this number?

Reviewer 2



NOTE: I am reviewing two papers that appear to be by the same authors and on the same general topic. The other one is on noiseless queries with additional side information. This paper gives upper and lower bounds on the query complexity of clustering based on noisy pairwise comparisons. While I am familiar with some of the related literature, my knowledge of it is far from complete, so it's a little hard to fully judge the value of the contributions. One limitation is that the main result would be extremely easy to obtain if the noise were independent when sampling the same pair multiple times. On the other hand, the case where the same noisy answer is always returned is well-motivated in certain applications. My overall impression of both papers is that the contributions seem to be good (if correct), but the writing could do with a lot of work. I acknowledge that the contributions are more important than the writing, but there are so many issues that my recommendation is still borderline. It would have been very useful to have another round of reviews where the suggestions are checked, but I understand this is not allowed for NIPS. Here are some of my more significant comments: - Many of the central results are not written precisely. The order of quantification is Lemma 1 is unclear (e.g., Pr(for all v) vs. for all v, Pr(...)). The last step in its proof only holds for sufficiently large k, whereas the lemma is stated for arbitrary k. The appendix is full of theorems/lemmas/claims with the imprecise phrase "with high probability", and in some cases it seems to be crucial to specify the convergence rate (even if it's done in the proof, it should still appear in the formal statement). - Near the bottom of p3, the runtime and query complexity are not consistent, with the latter exceeding the former in certain regimes like k = n^(1/6) or p very close to 1/2. I guess this is because certain dependences were omitted in the runtime (which they shouldn't be without further statement). - The appendices are often hard to follow. The strange and somewhat unclear phrases "Suppose if possible" and "Let if possible" are used throughout. Inequalities like the one at the end of Claim 2's proof appear questionable at first (e.g., when p is close to half), and the reader has to go back and find the definitions of c, c' (and check their dependence on p). There are a lot of statements along the lines of "similarly to [something earlier]", which I trusted but wasn't certain of. - The paper is FULL of grammar mistakes, typos, and strange wording. Things like capitalization, plurals, spacing, and articles (a/an/the) are often wrong. Commas are consistently mis-used (e.g., they usually should not appear after statements like "let" / "we have" / "this means" / "assume that" / "is", and very rarely need to be used just before an equation). Sentences shouldn't start with And/Or. There are typos like "entrees", "once crucial". The word "else" is used strangely (often "otherwise" would be better). This list of issues is far from complete -- regardless of the decision, I strongly urge the authors to proof-read the paper with as much care as possible. Here are some comments of a technical nature: - The paper is so dense that I couldn't check the upper bound proof in as much detail as ideal. I was reasonably convinced of the proof of the lower bound. - In Section 2, I don't see why the first "balanced" condition is useful. Even with a given maximum cluster size, the exact same problem as that described in the first few sentences can still occur, making the problem ill-posed. I think you should only keep the condition on the minimum cluster size. - The numerical section does not seem to be too valuable, mainly because it does not compare against any existing works or simple baselines. Are your methods really the only ones that can be considered/implemented? Some less significant comments: - The second half of the abstract can be considerably shortened - The algorithm descriptions are very wordy. It might have been useful to accompany them with figures and/or an "Algorithm" environment. - Sometimes extra citations are needed (e.g., runtime of heaviest weight subgraph, Turan's theorem, and maybe previous use of the TV->KL technique on p5 like Auer et al 1998 bandit paper) - I would remove "initiate [a rigorous theoretical study]" from the abstract/intro. Given that there are so many related works, I don't think this is suitable. It is even less suitable to simultaneously say it in two different papers. - In the formal problem statement "find Q subset of V x V" makes it sound like the algorithm is non-adaptive. - The sentence "which might also be the aggregated..." is confusing and should be removed or reworded - Many brackets are too small throughout the appendix - In the Unknown k algorithm, l (which should be \ell?) is never updated, so k=2l always assigns the same value - I would avoid phrases like "information theoretically optimal algorithm", which makes it sound like it is THE algorithm that minimizes the error probability. [POST-AUTHOR FEEDBACK COMMENTS] The authors clarified a few things in the responses, and I have updated the recommendation to acceptance. I still hope that a very careful revision is done for the final version considering the above comments. I can see that I was the only one of the 3 reviewers to have significant comments about the writing, but I am sure that some of these issues would make a difference to a lot more NIPS readers, particularly those coming from different academic backgrounds to the authors. Some specific comments: - The authors have the wrong idea if they feel that "issues like capitalization and misplaced commas" were deciding factors in the review. These were mentioned at the end as something that should be revised, not as a reason for the borderline recommendation. - I am still confused about the "balanced" condition and minimum/maximum cluster size. My understanding is that if you only assume a bound on the maximum size, then the problem is in general ill-posed, by your example (definitions of C_1 and C_2) on lines 149-150. If you only assume a bound on the minimum size, everything looks fine. You could also assume both simultaneously, but this doesn't seem to be what you are doing (since on line 155 you write "either of" rather than "both of"). Consider re-wording/explaining more as needed. - Please revise/re-word the runtime and query complexity on p3 -- if the stated runtime only holds for p = constant, then you should add something like "when p is constant". I think cases like k = n^(1/6) (where samples seems to exceed runtime) might still look strange to some readers, but since there is no true contradiction, I don't insist on anything. - It is important that theorems and lemmas are stated precisely (e.g., all assumptions are included like k being large or certain quantities behaving as constants, and convergence rates of probabilities are specified if they play an important role in the later analysis) - Although it is not a major point, I will mention that I don't recall seeing "Suppose if possible" or "Let if possible" despite reading mathematical papers from many domains. The latter in particular sounds very strange. Even for the former, at least for me, a re-wording to something like "Suppose by way of contradiction that..." would be clearer.